# DiffPano: Scalable and Consistent Text to Panorama Generation with Spherical Epipolar-Aware Diffusion

**Weicai Ye**[1,3,*]    **Chenhao Ji**[2,*]    **Zheng Chen**[4]    **Junyao Gao**[2]    **Xiaoshui Huang**[3]
Song-Hai Zhang[4]    Wanli Ouyang[3]    Tong He[3,✉]    Cairong Zhao[2,✉]    Guofeng Zhang[1,✉]
[1]State Key Lab of CAD&CG, Zhejiang University    [2]Tongji University
[3]Shanghai AI Laboratory [4]Tsinghua Univerisity
maikeyeweicai@gmail.com    jichenhao@tongji.edu.cn    zhangguofeng@zju.edu.cn

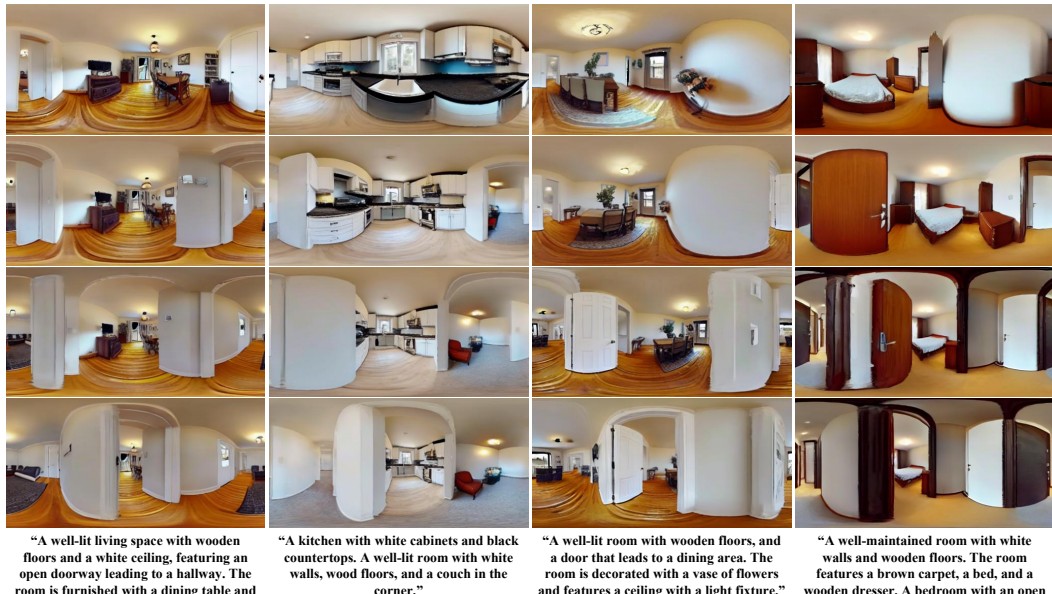

"A well-lit living space with wooden floors and a white ceiling, featuring an open doorway leading to a hallway. The room is furnished with a dining table and chairs. "

"A kitchen with white cabinets and black countertops. A well-lit room with white walls, wood floors, and a couch in the corner."

"A well-lit room with wooden floors, and a door that leads to a dining area. The room is decorated with a vase of flowers and features a ceiling with a light fixture."

"A well-maintained room with white walls and wooden floors. The room features a brown carpet, a bed, and a wooden dresser. A bedroom with an open door leading to the hallway."

Figure 1: **DiffPano allows scalable and consistent panorama generation (i.e. room switching) with given unseen text descriptions and camera poses.** Each column represents the generated multi-view panoramas, switching from one room to another.

## Abstract

Diffusion-based methods have achieved remarkable achievements in 2D image or 3D object generation, however, the generation of 3D scenes and even 360° images remains constrained, due to the limited number of scene datasets, the complexity of 3D scenes themselves, and the difficulty of generating consistent multi-view images. To address these issues, we first establish a large-scale panoramic video-text dataset containing millions of consecutive panoramic keyframes with corresponding panoramic depths, camera poses, and text descriptions. Then, we propose a novel text-driven panoramic generation framework, termed DiffPano, to achieve scalable, consistent, and diverse panoramic scene generation. Specifically, benefiting from the powerful generative capabilities of stable diffusion, we fine-tune a single-view text-to-panorama diffusion model with LoRA on the established panoramic video-text dataset. We further design a spherical epipolar-aware multi-view diffusion model to ensure the multi-view consistency of the generated panoramic images.

---

*: Equal Contribution. ✉: Corresponding Author.

38th Conference on Neural Information Processing Systems (NeurIPS 2024).

Extensive experiments demonstrate that DiffPano can generate scalable, consistent, and diverse panoramic images with given unseen text descriptions and camera poses. Code and dataset are available at `https://zju3dv.github.io/DiffPano`.

# 1 Introduction

Generating scenarios from the text descriptions that meet one's expectations is an imaginative and marvelous journey, which has many potential applications, such as VR roaming [62] for metaverse [67, 68, 69, 72], physical world simulation [2, 6, 14], embodied agents in scene navigation and manipulation [75], etc. With the advent of the AIGC era, several works [8, 14, 20, 29, 42, 43, 48, 51, 52] on object generation even scene generation has emerged. However, they generally only generate a series of perspective images, making it impossible to comprehensively simulate the entire environment for scene understanding [27, 19, 66] and reconstruction [3, 5, 65, 64, 30, 35, 34, 26, 21, 7, 12, 47, 49, 58]. Given this, some methods [70, 9] try to generate panoramas to solve these problems by taking advantage of the inherent characteristics of panoramic images, which can capture the surrounding environment with a single shot.

These methods can be roughly divided into four categories: 1) Directly single-view equirectangular projection (ERP) panorama generation [70]. However, its camera is immovable, making it incapable of scene exploration; 2) Multiple perspective views generation methods [20, 43, 48, 40] without considering multi-view panorama generation. 3) The inpainting solutions are based on the infinite expansion of a single perspective view, which lacks 3D awareness. Single scene optimization methods [53] with inpainting have no generalization ability and cost too much time for optimization. 4) Directly extending the multiple perspective views generation methods [43] to the ERP panorama, which is difficult to converge and results in poor multi-view consistency (see Fig. 5).

This paper aims to generate scalable and multi-view consistent panoramic images from text descriptions and camera poses (see Fig. 1) with many potential applications such as immersive VR roaming with unlimited scapes and preview for interior home design. However, achieving this goal is not trivial. To the best of our knowledge, there is currently a lack of rich and diverse panoramic datasets to meet the task of text-to-multi-view ERP panorama generation. To this end, we propose a novel panoramic video-text dataset and a generation framework suitable for the text-to-multi-view panorama generation task, advancing the development of this field. Specifically, we first establish a large-scale panoramic video-text dataset using Habitat Simulator [41] (see Sec. 3), which contains millions of panoramic keyframes and corresponding panoramic depths, camera poses, and text descriptions. Next, built upon the proposed dataset, we propose a generation framework for consistent multi-view ERP panorama generation (see Sec. 4), termed DiffPano. The DiffPano framework consists of a single-view text-to-panorama diffusion model (see Sec. 4.1) and a spherical epipolar-aware multi-view diffusion model (see Sec. 4.2). The single-view text-to-panorama diffusion model is obtained by fine-tuning the stable diffusion model [38] of perspective images using LoRA [18]. Considering that the single-view pano-based diffusion model cannot guarantee the consistency of generated multi-view panoramas with different camera poses, we derive a spherical epipolar constraint applicable to panoramic images, inspired by the perspective epipolar constraint. We then incorporated it as a spherical epipolar-aware attention module (see Sec. 4.2) into the multi-view panoramic diffusion model to ensure the multi-view consistency of the generated ERP panoramic images.

Since there are no related methods for comparison, we try to extend the MVDream method [43] to generate multi-view ERP panoramas. Trained and tested on the proposed panoramic video-text dataset, extensive experiments demonstrated that compared to the modified MVDream, our proposed multi-view panorama generation based on spherical epipolar-aware attention can generate more scalable and consistent panoramic images. Our method also demonstrates the generalization ability of the original diffusion model to generate satisfactory multi-view ERP panoramas with given unseen text descriptions and camera poses.

Our contributions can be summarized as follows: 1) To the best of our knowledge, we are the first to propose a scalable and consistent multi-view panorama generation task from text descriptions and camera poses. 2) We established a large-scale diverse and rich panoramic video-text dataset, which fosters the research of text-to-panoramic video generation. 3) We propose a novel text-driven panoramic generation framework with a spherical epipolar attention module, allowing scalable and consistent panorama generation with unseen text descriptions and camera poses.

## 2 Related work

### 2.1 Single-View Panorama Generation

Recently, latent diffusion model (LDM) methods have attracted widespread attention, and many single-view panorama generation works [70, 48, 74, 24, 60, 56, 33, 54, 62] have emerged, achieving remarkably impressive results. Among them, MVDiffusion [48] simultaneously generates eight fixed-viewpoint perspective images through a multi-view Correspondence-Aware diffusion model and stitches them together to produce a panorama. However, it cannot support the generation of top and bottom views, and the generated panorama resembles wide-angle images with an extensive field of view rather than true $360°$ images. Some methods [55, 13] solve this problem using equirectangular projection (ERP) and try to facilitate the interaction between the left and right sides during the panorama generation process to enhance the left-right continuity property inherent in ERP images. To address the domain gap between panorama and perspective images, PanFusion [70] proposed a novel dual-branch diffusion model that mitigates the distortion of perspective images projected on panoramas while providing global layout guidance. However, its more complex model architecture incurs longer inference times for panorama generation. In addition, PanFusion cannot be expanded as an effective pre-trained model to the multi-view panorama generation task due to its excessive network parameters. To strike a balance between computational complexity and ensuring left-right continuity of panoramas, our proposed single-view panorama-based stable diffusion model only requires fine-tuning with LoRA [18] to learn panoramic styles and achieve good edge continuity while maintaining higher generation speed and simpler architecture.

The existing single-view panorama generation methods cannot achieve scalable panorama generation. The core of our paper lies in the generation of multi-view consistent panoramic images, which we will introduce in Section 2.2. More importantly, the single-view panoramic image generated by previous methods mainly supports 3DoF roaming, while our method can generate multi-view panoramic images for 6DoF roaming, which can serve as the inputs for $360°$ Gaussian Splatting [23] or $360°$ NeRF [10, 11]. Our method also has a great potential value in $360°$ relightable novel view synthesis with the combination of $360°$ multi-view inverse rendering method [25].

### 2.2 Multi-View Image Generation

To the best of our knowledge, there is no work focusing on multi-view panorama generation. We review the existing works about multi-view generation for perspective images in this part.

Zero123 [29] laid the foundation for 3D object generation based on multi-view generation, while the pose-guided diffusion model [51, 40] explored consistent view synthesis of scenes. However, iteratively applying the diffusion model to generate individual views in the multi-view generation task may lead to poor multi-view consistency of generated images due to accumulated errors. To generate high-quality multi-view images simultaneously, some methods [31, 43, 57, 32] modify the UNets in the diffusion model into a multi-branch form and achieve the effect of generating consistent multi-view images through the interaction between different branch.

Currently, most multi-view generation tasks focus on generating multi-view perspective images of single objects [43, 61, 63, 28, 46, 45] or scenes [51, 15], while minimal research has been conducted on multi-view panorama generation. Narrow FoV (field-of-view) drawbacks of perspective images lead to the fact that the existing generation methods can only generate a very local region of the scene at a time. Our work focuses on the task of exploring the generation of $360°$ images from multiple different viewpoints. Due to the camera projection difference between panoramic and perspective images, achieving consistency in multi-view panoramas is challenging. It is impossible to directly apply the existing epipolar attention module [51, 20] to multi-view panoramas. We strive to derive the spherical epipolar line formula for panoramic images and propose a spherical epipolar attention module to ensure the multi-view consistency of the generated panoramas.

### 2.3 Panoramic Dataset

Great progress in text to single-view panorama generation has been witnessed. However, text-to-multi-view panorama generation is still a blank slate. One of the main limitations of this task is the lack of suitable datasets. The common panoramic datasets used in single-view panorama generation consist of indoor HDR dataset [16], outdoor HDR dataset [71], HDR360-UHD dataset [9], Structured3D [73],

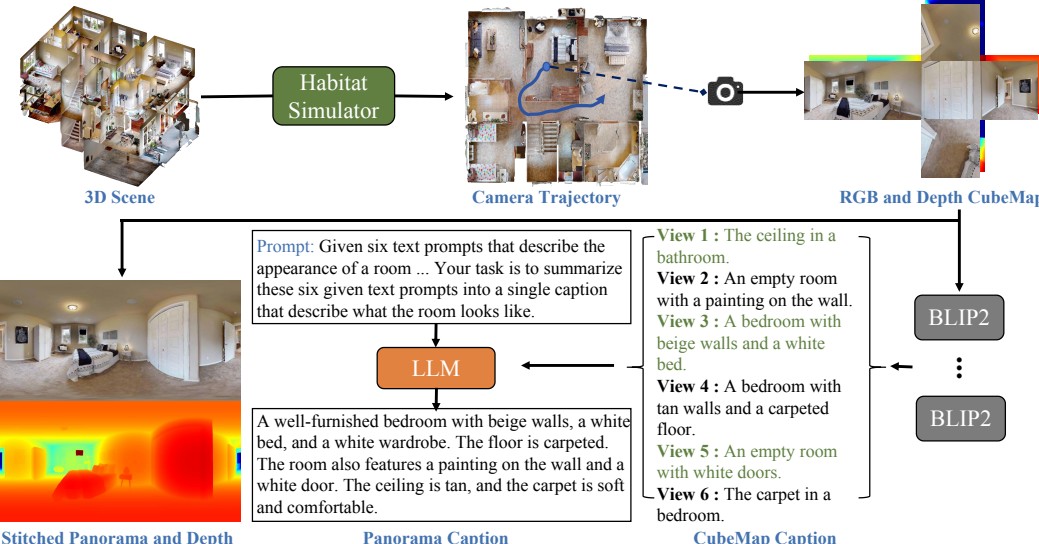

Figure 2: **Panoramic Video Construction and Caption Pipeline.**

Standford 2D-3D-S [1], and Matterport3D dataset [4], etc. Most of these datasets are relatively small in scale and only have single-view panoramas, which cannot support multi-view panorama generation, except Matterport3D [4]. In addition, the sky box images in Matterport3D [4] contain only sparse views. Although HM3D [37] provides the textured mesh of 1000 scenes, it lacks the corresponding text description for each view. To generate multi-view panoramas, we render cube maps at each viewpoint in the 3D meshes of HM3D, using the Habitat Simulator [41], and stitch them into panoramas. We generate complete text descriptions corresponding to the panoramas by using Blip2 [22] to create text descriptions for each face of the cube map separately, and then summarizing them using Llama2 [50]. In this way, we obtain a panoramic video-text dataset that includes camera poses, corresponding panoramas, and text descriptions of the panoramas, which facilitates subsequent multi-view panorama generation tasks.

## 3   Panoramic Video-Text Dataset

Due to the lack of high-quality panorama-text datasets, most text-to-panorama generation tasks require researchers to construct their own datasets. The dataset constructed in PanFusion [70] suffers from blurriness at the top and bottom of the panoramic images, and the corresponding text descriptions are not precise enough. To address these issues, we utilize the Habitat Simulator [41] to randomly select positions within the scenes of the Habitat Matterport 3D (HM3D) [37] dataset and render the six-face cube maps. These cube maps are then interpolated and stitched together to form panoramas so we can obtain panoramas with clear tops and bottoms. To generate more precise text descriptions for the panoramas, we first use BLIP2 [22] to generate corresponding text descriptions for each obtained cube map, and then employ Llama2 [50] to summarize and receive accurate and complete text descriptions. Furthermore, the Habitat Simulator allows us to render images based on camera trajectories within the HM3D scenes, enabling the creation of a dataset that simultaneously includes camera poses, panoramas, and corresponding text descriptions. This dataset will be utilized in the multi-view panorama generation (see Sec. 4.2).

## 4   Proposed Method: DiffPano

DiffPano is capable of generating multi-view consistent panoramas conditioned on camera viewpoints and textual descriptions, as illustrated in Fig. 1. In this section, we first introduce our single-view panorama stable diffusion in Sec. 4.1. We then elaborate on how to extend single-view panorama generation to multi-view consistent panorama generation by leveraging the spherical epipolar attention module in Sec. 4.2.

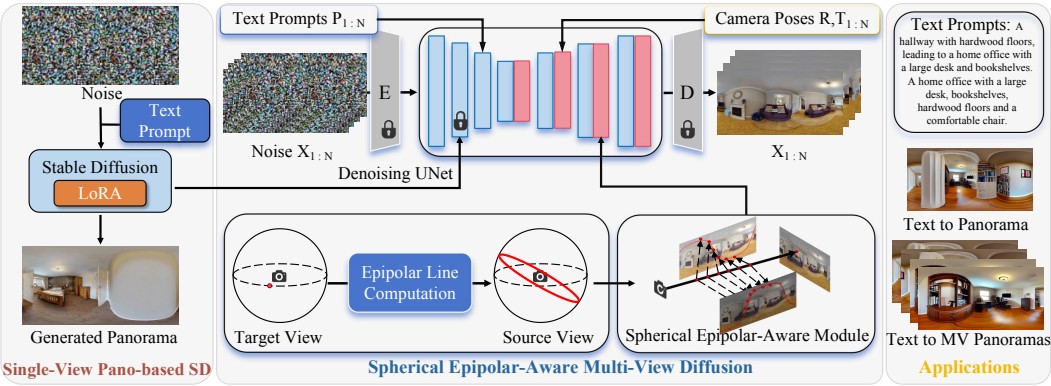

Figure 3: **DiffPano Framework.** The DiffPano framework consists of a single-view text-to-panorama diffusion model and a spherical epipolar-aware multi-view diffusion model. It can support text to single-view panorama or multi-view panorama generation.

## 4.1 Single-View Panorama-Based Stable Diffusion

A straightforward way to generate a single-view panorama from text is to train a text-to-panorama diffusion model from scratch with a large number of text-panorama pairs, which is both time-consuming and computationally expensive. However, stable diffusion [38] leverages a vast amount of perspective images and their corresponding textual descriptions as training data, endowing it with excellent prior knowledge of perspective images and strong text understanding capabilities. An economical and effective way for panorama generation is to fine-tune the trained perspective diffusion model with a few text-panorama pairs. To this end, panorama generation from text can be regarded as a style transfer of images generated by stable diffusion, converting them from perspective style to panoramic style, and requiring them to satisfy the left-right continuity property of panoramas.

**LoRA-based fine-tuning**   Diffusion models for text-to-image generations possess excellent prior knowledge of 2D images and strong text comprehension capabilities. We aim to preserve these abilities of the model while fine-tuning it to generate images in the style of panoramas. We employ the Low-Rank Adaptation (LoRA) [18] fine-tuning method, which has been previously used in large language models. Compared to full fine-tuning, LoRA is faster and requires fewer computational resources. In our approach, we freeze all the parameters of the original Stable Diffusion model and add trainable layers to the UNet component using the LoRA fine-tuning method. We then train the model using our custom-created panorama-text dataset. To improve the left-right continuity of the generated images, we perform data augmentation on the panorama training dataset by randomly concatenating a portion of the right side of the panorama to the left side. Experiments demonstrate that the panoramas generated using this method exhibit satisfactory left-right continuity.

## 4.2 Spherical Epipolar-Aware Multi-View Diffusion

Built upon our proposed single-view panorama stable diffusion in Sec. 4.1, we extend the single-view diffusion model to a multi-view diffusion model with a spherical epipolar-aware attention module to generate multi-view scalable and consistent panoramas.

**Spherical Epipolar-Aware Attention Module**   Epipolar attention was proposed in [20, 51] to ensure consistency between generated multi-view perspective images. However, due to the differences in imaging methods between perspective and panoramic views, existing epipolar attention cannot be directly used for panoramic views. To overcome this challenge, we derived the epipolar line for panoramic images in the equirectangular projection (ERP), and the specific proof process is provided in Appendix A. Equation 14 shows the mathematical form of the spherical epipolar line in ERP images. The spherical epipolar line is visualized in the spherical epipolar-aware attention module of Fig. 3. We extend the principle of epipolar attention to panoramic images to implement the spherical epipolar-aware attention module. Given a pixel $\mathbf{p}$ in the target view, we calculate its corresponding spherical coordinates $\mathbf{p}_{sphere}$ based on the spherical projection process:

$$\theta = (0.5 - \frac{x_{pix}}{W}) \cdot 2\pi$$
$$\phi = (0.5 - \frac{y_{pix}}{H}) \cdot \pi, \tag{1}$$

where $x_{pix}$ and $y_{pix}$ are the pixel coordinates of $\mathbf{p}$, $\theta$ and $\phi$ are its corresponding spherical coordinates and $W$ and $H$ are the resolutions of panorama. We then transform the spherical coordinate system to the Cartesian coordinate system to obtain the camera coordinates $p_{camera}$ corresponding to $\mathbf{p}$ :

$$x_{cam} = \cos(\phi) \cdot \sin(\theta)$$
$$y_{cam} = \sin(\phi) \tag{2}$$
$$z_{cam} = \cos(\phi) \cdot \cos(\theta).$$

The camera coordinates of $\mathbf{p}_{cam}$ are converted to world coordinates $\mathbf{p}_{world}$ through the camera's pose matrix. This allows us to compute the ray from the camera center to $\mathbf{p}_{world}$ in the world coordinate system and construct the spherical epipolar attention module.

Given $N$ feature maps $F = F_i | 1 \leq i \leq N$ corresponding to $N$ panoramic images and their respective camera pose matrices, we implement cross-attention between different views through the spherical epipolar-aware module. During the generation process, each feature map in $F$ can be considered as the target view, and the $K$ nearest views are selected from the remaining features as reference views.

For each feature point in the target view feature map, we uniformly sample $S$ points on the ray between the feature point and the camera. All sampled points are reprojected onto the feature maps of the $K$ reference views, and the corresponding feature values are obtained through interpolation. We denote the features in the target view as $q$, and the features of all sampled points in the reference views as $k$ and $v$. The cross-attention is then constructed using these features.

Let $p_i$ be a feature point in the target view feature map $F_t$, and $p_{i,j} | 1 \leq j \leq S$ be the $S$ uniformly sampled points on the ray between $p_i$ and the camera center. We reproject these points onto the $K$ reference view feature maps $F_{r_k} | 1 \leq k \leq K$ to obtain the corresponding feature values $f_{i,j,k} | 1 \leq j \leq S, 1 \leq k \leq K$. The query $q_i$, key $k_i$, and value $v_i$ for the cross-attention mechanism are defined as follows:

$$q_i = F_t(p_i), \quad k_i = f_{i,j,k} | 1 \leq j \leq S, 1 \leq k \leq K, \quad v_i = f_{i,j,k} | 1 \leq j \leq S, 1 \leq k \leq K. \tag{3}$$

The cross-attention output $o_i$ for the feature point $p_i$ is computed as:

$$o_i = Attention(q_i, k_i, v_i) = softmax(\frac{q_i k_i^T}{\sqrt{d}}) v_i, \tag{4}$$

where $d$ is the dimension of the query and key vectors.

**Positional Encoding** To enhance the model's understanding of 3D spatial relationships between different views, we follow EpiDiff [20] to employ the positional encoding method from Light Field Networks (LFN) [44]. In the world coordinate system, let $p_i$ be a pixel in the target view, and $o_i$ and $d_i$ be the origin and direction of the ray between $p_i$ and the camera center, respectively. The Plücker coordinates $r_i$ of the ray are computed as:

$$r_i = (o_i \times d_i, d_i). \tag{5}$$

For each sampled point $p_{i,j}$ on the ray, its corresponding spherical depth $z_{i,j}$ is transformed using a harmonic transformation to get $\gamma_z(z_{i,j})$. Similarly, the Plücker coordinates $r_i$ are transformed as $\gamma_r(r_i)$. The positionally encoded features $\gamma_r(r_i)$ and $\gamma_z(z_{i,j})$ are then concatenated to obtain the combined positional encoding $\gamma(r_i, z_{i,j})$:

$$\gamma(r_i, z_{i,j}) = [\gamma_r(r_i), \gamma_z(z_{i,j})]. \tag{6}$$

The combined positional encoding $\gamma(r_i, z_{i,j})$ is then concatenated with the feature maps $F_t$ and $F_{r_k} | 1 \leq k \leq K$ to obtain the enhanced feature representations $\hat{F}t$ and $\hat{F}r_k | 1 \leq k \leq K$:

$$\hat{F}t(p_i) = [F_t(p_i), \gamma(r_i, z_{i,j})], \quad \hat{F}r_k(p_{i,j}) = [F_{r_k}(p_{i,j}), \gamma(r_i, z_{i,j})], \tag{7}$$

where $[\cdot, \cdot]$ denotes concatenation. These enhanced feature representations are then used in the cross-attention mechanism to improve the model's understanding of 3D spatial relationships between different views.

**Two-Stage Training** The main difference between panoramic images and perspective images is that panoramic images contain $360°$ content of the surroundings, while perspective images only contain content from a given viewpoint. Therefore, when the camera only rotates or translates by a small amount, the corresponding content in the panoramic image hardly changes. To make the generated multi-view panoramic images better match each corresponding text, we divide the training into two stages. In the first stage, we use the selected dataset with almost no change in image content (small camera movement) for training, which enhances the effect of the spherical epipolar-aware attention module. In the second stage, we increase the camera movement distance between each viewpoint and train with images that generate new content, improving the model's ability to understand text based on changes in perceived spatial location while ensuring multi-view consistency and enhancing scalability.

# 5 Experiment

**Dataset** We leverage the Habitat Simulator [41] to render a panoramic video dataset based on the Habitat Matterport 3D (HM3D) dataset [37]. The pipeline of dataset rendering and captioning is shown in Sec. 3. After post-processing such as dataset filtering, we constructed 8,508 panorama-text pairs as training sets for single-view panorama generation. For multi-view panorama generation, we constructed 19,739 multi-view panorama-text pairs with nearly identical image content and 18,704 multi-view panorama-text pairs with different image contents as training sets. For specific details regarding the dataset, please refer to Appendix B.

**Implementation Details** In the multi-view panorama generation, we simultaneously generate $N = 4$ panoramas from different viewpoints. Within the spherical epipolar-aware attention module, we consider the two nearest views to the target view as reference views, i.e., $K = 3$, and sample $S = 10$ points along each ray. We conducted separate training for 100 epochs on datasets with almost identical image content and datasets with varying image content. Please refer to Appendix C for further implementation details.

**Evaluation Metrics** To evaluate the performance of our proposed single-view panorama-based stable diffusion model, we employ three commonly used metrics: Fréchet Inception Distance (FID) [17], Inception Score (IS) [39], and CLIP Score (CS) [36]. FID measures the similarity between the distribution of generated panoramas and the distribution of real images. IS assesses the quality and diversity of the generated panoramas. CS is utilized to evaluate the consistency between the input text and the generated panoramas. To further evaluate the consistency of multi-view panorama generation, we leverage the Peak Signal-to-Noise Ratio (PSNR) and Structural Similarity Index Measure (SSIM) [59] metrics. PSNR and SSIM quantify the pixel-level differences and structural similarity between the generated views respectively.

## 5.1 Single-View Panorama Generation

**Baselines** We evaluate the performance of our proposed method by comparing it with the following baseline approaches for text-to-panorama generation:

- *Text2Light* [9]is a two-stage approach that first generates a low-resolution panorama based on the input text, and then expands it to ultra-high resolution.

- *PanFusion* [70]is a dual-branch text-to-panorama model that aims to mitigate the distortion caused by projecting perspective images onto a panoramic canvas while providing global layout guidance.

Since the panoramic images generated by MVDiffusion [48] do not include top and bottom viewpoints, they are not complete panoramas. Therefore, we do not compare our method with MVDiffusion.

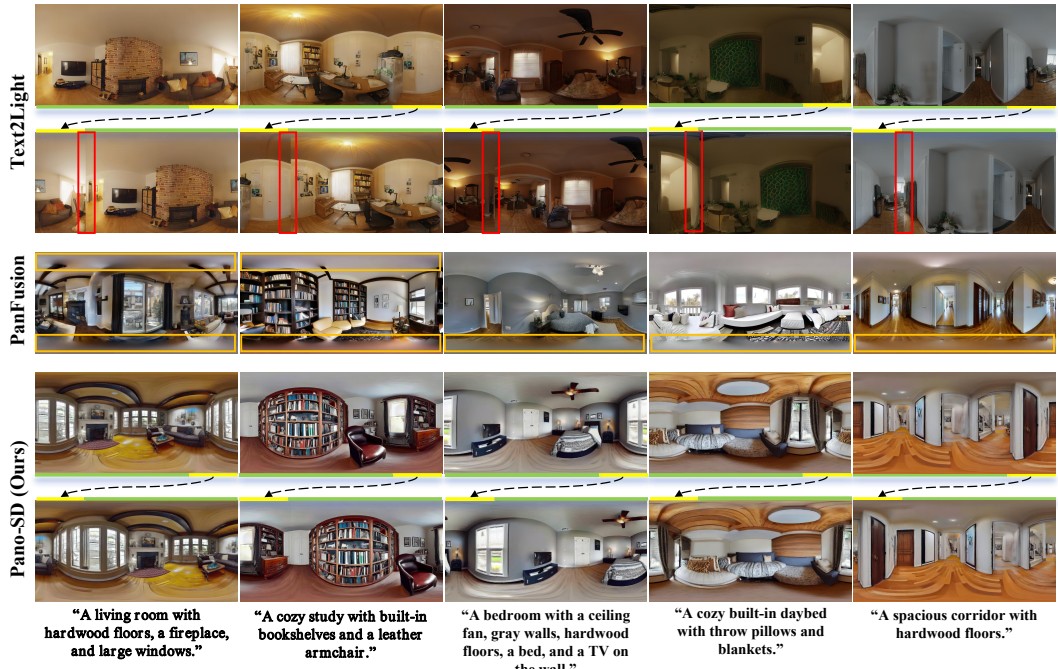

Figure 4: **Text to Panorama Comparison between TextLight, PanFusion, and Ours.**

Table 1: Quantitative Panorama Comparisons with Baseline Methods

| Method | FID↓ | IS↑ | CS↑ | Inference time(s)↓ |
|---|---|---|---|---|
| Text2Light | 76.50 | 3.60 | 27.48 | 50.4 |
| PanFusion | **47.62** | **4.49** | 28.73 | 27.6 |
| Pano-SD(Ours) | 48.52 | 3.30 | **28.98** | **5.1** |

**Quantitative Results**    Table 1 presents the quantitative results. The FID, IS, and CS values of Text2Light are from the PanFusion [70]. We trained our proposed Pano-SD using the same dataset as PanFusion [70] and re-evaluated the performance of PanFusion. From the results, it can be observed that our method slightly outperforms others in terms of CS value and achieves a significantly lower FID value compared to Text2Light, which is close to PanFusion [70]. Moreover, due to the simplicity and efficiency of our model architecture, our method has a substantial advantage in inference time, which can significantly improve the efficiency of subsequent multi-view generation tasks.

**Qualitative Results**    The qualitative comparison results are shown in Figure 4. We compare our method with the two models trained on their respective datasets. The panoramas generated by Text2Light exhibit poor left-right consistency. On the other hand, the panoramas generated by PanFusion suffer from blurriness at the bottom and top regions, which affects the overall integrity of the panoramas. In contrast, our model is capable of generating panoramas with clear bottom and top regions and better left-right continuity. However, due to the quality of the dataset, the image quality may be slightly inferior.

## 5.2    Multi-View Panorama Generation

To the best of our knowledge, there is no method for multi-view panorama generation, the existing SOTA method MVDream for perspective images cannot be directly applied to multi-view panorama generation. To verify the validity of our proposed spherical epipolar-aware attention module, we adapted MVDream to the panorama generation task as a comparative baseline.

Specifically, we remove the spherical epipolar-aware attention module from our method and load the pre-trained LoRA layers from Pano-SD. We then convert the 3D self-attention in the UNet to the form used in MVDream [43]. Additionally, we transform the camera pose matrix into camera

Table 2: User Study of Text to Multi-view Panoramas

| Method | Image quality↑ | Image-text consistency↑ | Multi-view consistency↑ |
|---|---|---|---|
| MVDream | 3.6 | 3.7 | 3.8 |
| PanFusion | 3.8 | 4.0 | 3.0 |
| DiffPano(Ours) | **4.0** | **4.2** | **4.3** |

Table 3: Ablation Study on the Number of Sampling Points and Reference Frames

| | FID↓ | IS↑ | CS↑ | PSNR↑ | SSIM↑ | Inference time(s)↓ |
|---|---|---|---|---|---|---|
| $K = 1, S = 10$ | 73.30 | 3.40 | 22.14 | 29.99 | 0.69 | 30.06 |
| $K = 2, S = 10$ | 66.02 | 3.34 | 22.92 | 32.04 | 0.79 | 33.12 |
| $K = 3, S = 10$ | 69.89 | 3.58 | 22.76 | 32.29 | 0.81 | 35.79 |
| $K = 4, S = 6$ | 68.39 | 3.57 | 22.74 | 33.32 | 0.86 | 26.61 |
| $K = 4, S = 8$ | 67.30 | 3.54 | 22.66 | 33.00 | 0.84 | 32.23 |
| $K = 4, S = 10$ | 65.98 | 3.37 | 22.59 | 33.39 | 0.87 | 37.91 |
| $K = 4, S = 12$ | 62.79 | 3.26 | 22.65 | 32.89 | 0.83 | 43.72 |

embeddings through a 2-layer MLP and add it as a residual to the time embeddings. The qualitative comparison of the experiment is shown in Fig 5. Under the same training iteration, DiffPano significantly outperforms MVDream [43] in terms of multi-view consistency. Our method can maintain consistency in the details of multi-view images, while MVDream can only achieve a certain level of similarity in the overall images. Even compared to MVDream with twice the number of training iterations, our method still performs better in terms of consistency.

**User Study** We collected 20 text prompts and recruited nearly 50 volunteers to evaluate text-to-multi-view panoramic image generation. Evaluation metrics of multi-view ERP panorama generation include the quality of multi-view panoramic images, multi-view consistency, and the consistency between the text and multi-view panoramas. Experimental results on Tab. 2 show that our method can generate multi-view panoramic images with better quality, higher text and image similarity, and more consistent multi-view images, compared with MVDream [43] and PanFusion [70]. See more qualitative results in Fig. 6 to Fig. 10.

## 5.3 Ablation Study

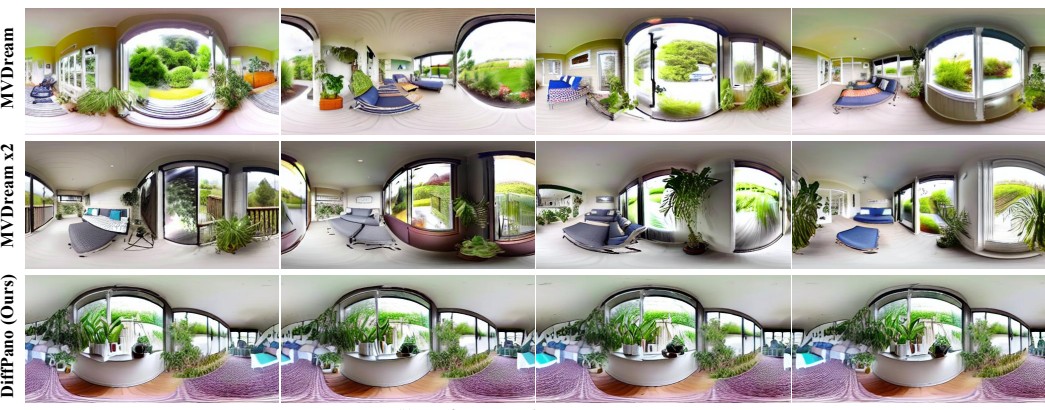

"A peaceful sunroom with a daybed, plants, and floor-to-ceiling windows."

Figure 5: **Comparisons with MVDream.** DiffPano can generate more consistent multi-view panoramas. "MVDream×2" denotes MVDream is trained with twice iteration number relative to our method.

Table 4: Ablation Study of One-Stage vs Two-Stage Training

|  | FID↓ | IS↑ | LPIPS↓ | PSNR↑ | SSIM↑ |
|---|---|---|---|---|---|
| One-stage | 82.92 | 4.52 | 0.0454 | 31.64 | 0.74 |
| Two-stage | 74.08 | 3.13 | 0.0610 | 31.54 | 0.73 |

**Number of Reference Views and Sample Points**    To explore the influence of varying numbers of reference frames and sampling points on model performance, we generate 5-frame multi-view panoramas simultaneously, and compute the FID, IS, and CS metrics for the first frame of each group to assess the quality of the generated panorama under different quantities of reference frames and sampling points. Furthermore, we set the same camera pose for the first and last frames of each generated panorama group, and calculate the PSNR and SSIM values between these two frames to evaluate the model's multi-view consistency. As shown in Table. 3, increasing the number of reference frames and sampling points can improve the quality of generated panoramas to a certain extent, but the changes in image diversity and consistency with text remain marginal. With an increasing number of reference frames and sampling points, the model's consistency exhibits improvement, however, when the number of sampling points becomes excessive ($S$=12), the multi-view consistency of the model diminishes.

**One-Stage vs Two-Stage**    We conduct ablation experiments on one-stage and two-stage training. Table. 4 shows that the two-stage method will obtain the higher FID values. The IS of the two-stage training method is lower, and the diversity is reduced to a certain extent, which is slightly worse than the one-stage training. However, it should be noted that the images after one-stage training will have ghosting, but the two-stage will not.

## 6    Conclusion

We have proposed the panoramic video-text dataset and panorama generation framework with spherical epipolar-aware attention for text-to-single-view or multi-view panorama generation. Extensive experiments demonstrate that our method can achieve scalable, consistent, and diverse multi-view panoramas.

**Limitation and Future Work**    Although our method demonstrates the ability to generate consistent multi-view panoramas under the same setting as the training phase, it is important to note that as the number of frames increases during inference, the model tends to hallucinate content.

Exploring the use of video diffusion models to improve the consistency of generated multi-view panoramas is a promising direction. Longer panoramic videos are expected to be realized based on the generated panoramas as conditions.

## 7    Acknowledgements

This work was partially supported by the National Key Research and Development Program of China (No. 2023YFF0905104) and National Natural Science Foundation of China (Nos. 61932003, 62076184 and 62473286).

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

# A  Spherical Epipolar Line Computation

Given the pixel coordinates of a point in the panorama of the target view $i$, the corresponding epipolar line in the panorama of the source view $j$ can be calculated. First, according to Eq. ( 1), the camera coordinates $\mathbf{p}$ in the target view $i$ are computed. Based on the relative pose $\left\{\mathbf{R}^{i \to j}, \mathbf{T}^{i \to j}\right\}$, $\mathbf{p}'$ in the source view $j$ are calculated:

$$\mathbf{p}' = \mathbf{R}^{i \to j}\mathbf{p} + \mathbf{T}^{i \to j}. \tag{8}$$

Simultaneously, the coordinates $\mathbf{o}'$ of the camera origin $\mathbf{o} = (0,0,0)^T$ in the target view $i$ projected onto the source view $j$ are computed:

$$\mathbf{o}' = \mathbf{R}^{i \to j}\mathbf{o} + \mathbf{T}^{i \to j}. \tag{9}$$

Then, we need to find the plane $L$ containing the three points $\mathbf{p}'$, $\mathbf{o}$, and $\mathbf{o}'$ in the camera coordinate system of the source view $j$. The intersection of this plane with the spherical surface is the desired epipolar line. The equation corresponding to plane $L$ is:

$$AX + BY + CZ + D = 0. \tag{10}$$

Substituting the coordinates of the three points into the equation yields the coefficients $A$, $B$, $C$, and $D$:

$$\begin{aligned} A &= \frac{z_{\mathbf{o}'} \cdot y_{\mathbf{p}'} - z_{\mathbf{p}'} \cdot y_{\mathbf{o}'}}{x_{\mathbf{p}'} \cdot y_{\mathbf{o}'} - x_{\mathbf{o}'} \cdot y_{\mathbf{p}'}} \cdot C \\ B &= \frac{z_{\mathbf{o}'} \cdot x_{\mathbf{p}'} - z_{\mathbf{p}'} \cdot x_{\mathbf{o}'}}{y_{\mathbf{p}'} \cdot x_{\mathbf{o}'} - y_{\mathbf{o}'} \cdot x_{\mathbf{p}'}} \cdot C \\ D &= 0. \end{aligned} \tag{11}$$

To simplify the representation of the equation for plane $L$, we introduce new coefficients $a_1$ and $a_2$:

$$\begin{aligned} a_1 &= \frac{z_{\mathbf{o}'} \cdot y_{\mathbf{p}'} - z_{\mathbf{p}'} \cdot y_{\mathbf{o}'}}{x_{\mathbf{p}'} \cdot y_{\mathbf{o}'} - x_{\mathbf{o}'} \cdot y_{\mathbf{p}'}} \\ a_2 &= \frac{z_{\mathbf{o}'} \cdot x_{\mathbf{p}'} - z_{\mathbf{p}'} \cdot x_{\mathbf{o}'}}{y_{\mathbf{p}'} \cdot x_{\mathbf{o}'} - y_{\mathbf{o}'} \cdot x_{\mathbf{p}'}}, \end{aligned} \tag{12}$$

the equation of plane $L$ can be simplified to:

$$a_1 X + a_2 Y + Z = 0. \tag{13}$$

According to Eq.1 and Eq.2, the epipolar line equation in the source view $j$ can be obtained:

$$y_{pix} = H \cdot \left[ \arctan\left( \frac{a_1 \sin\left(\frac{2\pi x_{pix}}{W}\right) - \cos\left(\frac{2\pi x_{pix}}{W}\right)}{a_2} \right) / \pi + 0.5 \right], \tag{14}$$

where $x_{pix}$ and $y_{pix}$ are the corresponding pixel coordinates.

# B  Experiment Details

**Dataset Processing**  In single-view panorama generation, we select 100 scenes from HM3D [37] and render cube maps from random viewpoints within each scene, which are then stitched into panoramas through interpolation. However, due to the imperfect quality of the corresponding scene meshes, which may have missing parts, we filter out images with a high proportion of zero-depth values based on their corresponding depth maps, ultimately creating a dataset of 8,508 panoramas.

Table 5: Quantitative Perspective Images Comparisons with Baseline Methods

| Method | FID↓ | IS↑ |
|---|---|---|
| MVDiffusion | 188.16 | 1.82 |
| PanFusion | 213.19 | **2.61** |
| Pano-SD(Ours) | **164.75** | 1.96 |

In the multi-view panorama generation task, we render images based on camera trajectories across scenes. To accommodate the first stage of training, we select camera trajectories with minimal changes. In contrast, for the second stage, the dataset comprises panoramas with more significant camera displacements between consecutive frames. Each training dataset consists of multiple panoramas, their corresponding pose matrices, and text descriptions. To construct a multi-view panorama dataset with nearly identical image content, we select the first few frames from each camera trajectory, as the camera movement between these frames is minimal and they are captured within the same scene. For datasets where the image content changes, the construction needs to be performed across the entire camera trajectory. By projecting the panorama from the subsequent frame onto the viewpoint of the previous frame using spherical projection, we compare the pixel value differences between the two. If more than 40% of the pixel values differ, we consider new content to have been generated between the two frames. After filtering, there are 19,739 data sets with nearly identical image content and 18,704 data sets with differing image content.

**Compare with Baseline Methods**

- *Text2Light* [9]: In the quantitative analysis, the FID, IS, and CS values are referenced from the results reported in PanFusion[70], which were obtained by testing on the same dataset after training. For the qualitative analysis, we directly employ their jointly trained model on both indoor and outdoor datasets to generate LDR (low dynamic range) panoramas based on text descriptions.

- *PanFusion* [70]: To compare the quality of the generated panoramas, we also train the model using panoramas stitched from MP3D skybox images. We employ the same dataset split and text descriptions as in [70], which includes 9,820 panoramas for training and 1,092 for evaluation. In the qualitative analysis, we use our own dataset constructed from HM3D[37]. The panoramas in our dataset do not suffer from blurriness at the bottom and top regions, but the overall image quality may be slightly lower.

**Compare with MVDream**    We adopt the method from MVDream[43] by formatting the $B \times N \times H \times W \times C$ tensor input to the self-attention module as $B \times NHW \times C$. This allows the model to simultaneously integrate features from all viewpoints during the self-attention computation, thereby improving the consistency among the generated multi-view panoramas. However, since MVDream is designed for multi-view image generation of objects, the camera pose matrices used in their method only contain rotation information without translation. In contrast, when generating camera embeddings, we use pose matrices that include both rotation and translation information, which increases the learning difficulty for the model. This may be one of the reasons why using this method does not yield particularly ideal results.

**Transform Panoramas to Perspective Views**    We converted the generated panoramas into perspective images and conducted quantitative comparisons, shown in Tab. 5. Experiments show that our method achieves the lowest FID, while our method is higher than MVDiffusion in IS and slightly lower than PanFusion. It should be noted that MVDiffusion directly generates perspective images and then stitches them into panoramas. It is not in the ERP format and does not have the top and bottom parts. Our panorama generation speed is faster.

### B.1 Applications

#### B.1.1 Text to Single-View Panorama

#### B.1.2 Text to Multi-View Panoramas

DiffPano can generate panoramic video frames with large camera pose spans and multi-view consistency based on diverse textual descriptions, thus achieving the effect of text-to-panoramic video, as shown in the Fig.11.

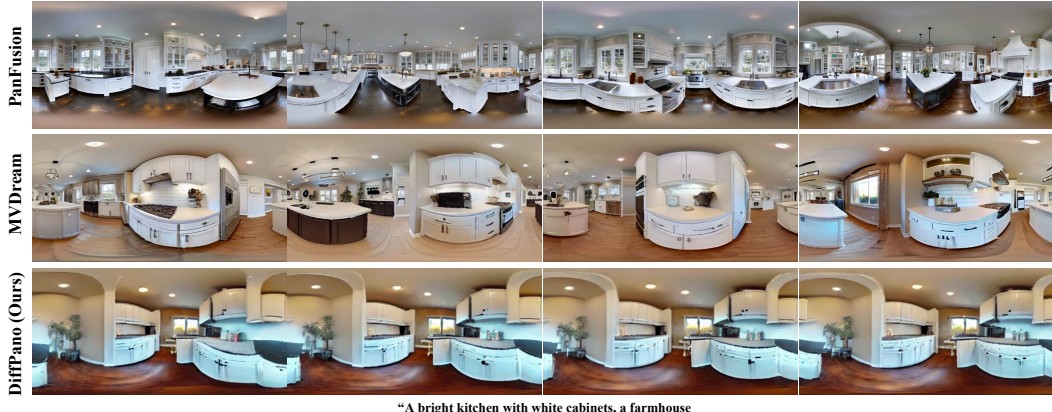

"A bright kitchen with white cabinets, a farmhouse sink, and a large island."

Figure 6: **Qualitative Comparisons of Text to Panoramic Videos.** Ours vs MVDream vs PanFusion.

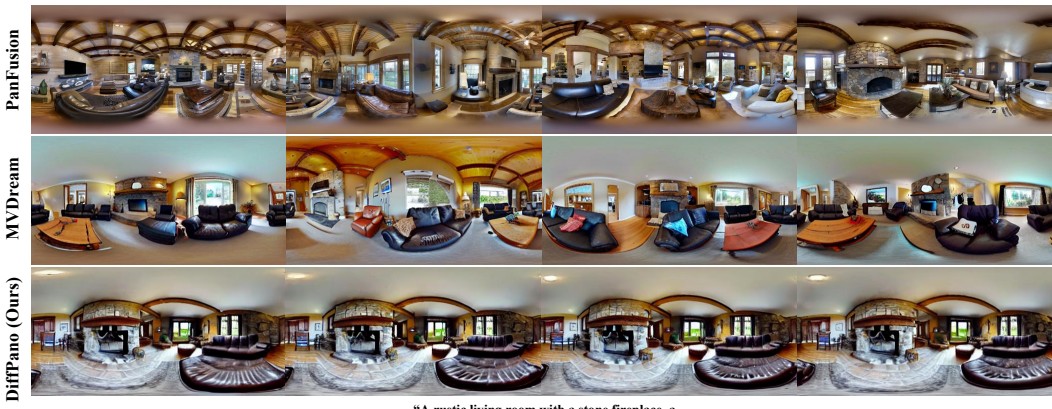

"A rustic living room with a stone fireplace, a leather sofa, and a wooden coffee table."

Figure 7: **Qualitative Comparisons of Text to Panoramic Videos.** Ours vs MVDream vs PanFusion.

## C Network Architecture and Training Details

**Network Architecture**    Our LoRA-based single-view panorama generation model produces panoramas with a resolution of 512×1024. In the multi-view panorama generation approach, we generate continuous frames of panoramas with a resolution of 256×512. Generating multiple frames of 512×1024 resolution panoramas simultaneously would consume a significant amount of computational resources. Moreover, our experiments reveal that generating multi-frame high-resolution panoramas requires an exceptionally long training time to improve the quality of the generated images. The network architecture of our multi-view panorama generation model is shown in Table 6 and Table7.

**Training Details**    We fine-tuned the Stable Diffusion v1.5 model using the LoRA method for single-view pano-based synthesis. The training was conducted on 6 A100 GPUs with 80GB memory for 100 epochs (approximately 6.5 hours), with a learning rate of 1e-4 and a batch size of 6. For

Table 6: Network architecture of DiffPano-1

| | Layer | Output | Additional Inputs |
|---|---|---|---|
| (1) | Latent Map | $4 \times 32 \times 64$ | |
| (2) | Conv. | $320 \times 32 \times 64$ | |
| | CrossAttnDownBlock1 | | |
| (3) | ResBlock | $320 \times 32 \times 64$ | Time emb. |
| (4) | AttnBlock | $320 \times 32 \times 64$ | Prompt emb. |
| (5) | ResBlock | $320 \times 32 \times 64$ | Time emb. |
| (6) | AttnBlock | $320 \times 32 \times 64$ | Prompt emb. |
| (7) | DownSampler | $320 \times 16 \times 32$ | |
| | CrossAttnDownBlock2 | | |
| (8) | ResBlock | $640 \times 16 \times 32$ | Time emb. |
| (9) | AttnBlock | $640 \times 16 \times 32$ | Prompt emb. |
| (10) | ResBlock | $640 \times 16 \times 32$ | Time emb. |
| (11) | AttnBlock | $640 \times 16 \times 32$ | Prompt emb. |
| (12) | DownSampler | $640 \times 8 \times 16$ | |
| | CrossAttnDownBlock3 | | |
| (13) | ResBlock | $1280 \times 8 \times 16$ | Time emb. |
| (14) | AttnBlock | $1280 \times 8 \times 16$ | Prompt emb. |
| (15) | ResBlock | $1280 \times 8 \times 16$ | Time emb. |
| (16) | AttnBlock | $1280 \times 8 \times 16$ | Prompt emb. |
| (17) | DownSampler | $1280 \times 4 \times 8$ | |
| | DownBlock | | |
| (18) | ResBlock | $1280 \times 4 \times 8$ | Time emb. |
| (19) | ResBlock | $1280 \times 4 \times 8$ | Time emb. |
| | MidBlock | | |
| (20) | ResBlock | $1280 \times 4 \times 8$ | Time emb. |
| **(21)** | **EAModule** | $\mathbf{1280 \times 4 \times 8}$ | |
| (22) | AttnBlock | $1280 \times 4 \times 8$ | Prompt emb. |
| (23) | ResBlock | $1280 \times 4 \times 8$ | Time emb. |
| | UpBlock | | |
| (24) | ResBlock | $1280 \times 4 \times 8$ | (19), Time emb. |
| (25) | ResBlock | $1280 \times 4 \times 8$ | (18), Time emb. |
| (26) | ResBlock | $1280 \times 4 \times 8$ | (17), Time emb. |
| **(27)** | **EAModule** | $\mathbf{1280 \times 4 \times 8}$ | |
| (28) | UpSampler | $1280 \times 8 \times 16$ | |
| | CrossAttnUpBlock1 | | |
| (29) | ResBlock | $1280 \times 8 \times 16$ | (16), Time emb. |
| (30) | AttnBlock | $1280 \times 8 \times 16$ | Prompt emb. |
| (31) | ResBlock | $1280 \times 8 \times 16$ | (14), Time emb. |
| (32) | AttnBlock | $1280 \times 8 \times 16$ | Prompt emb. |
| (33) | ResBlock | $1280 \times 8 \times 16$ | (12), Time emb. |
| (34) | AttnBlock | $1280 \times 8 \times 16$ | Prompt emb. |
| **(35)** | **EAModule** | $\mathbf{1280 \times 8 \times 16}$ | |
| (36) | UpSampler | $1280 \times 16 \times 32$ | |

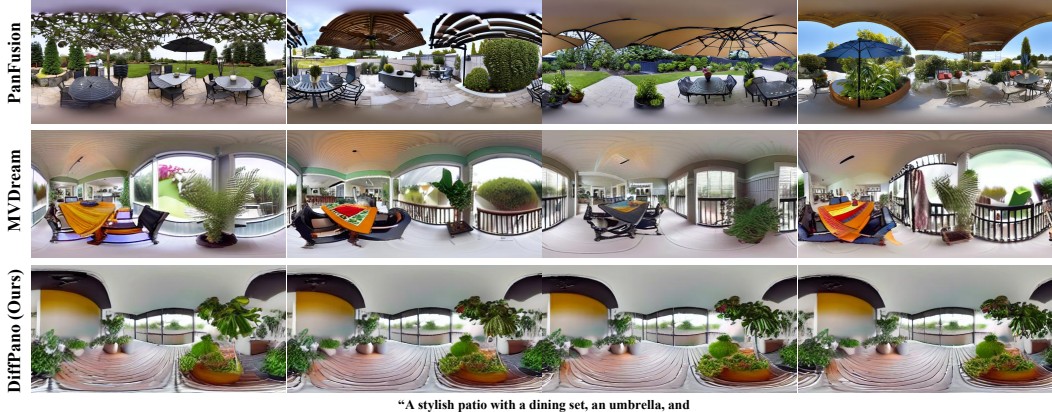

Figure 8: **Qualitative Comparisons of Text to Panoramic Videos.** Ours vs MVDream vs PanFusion.

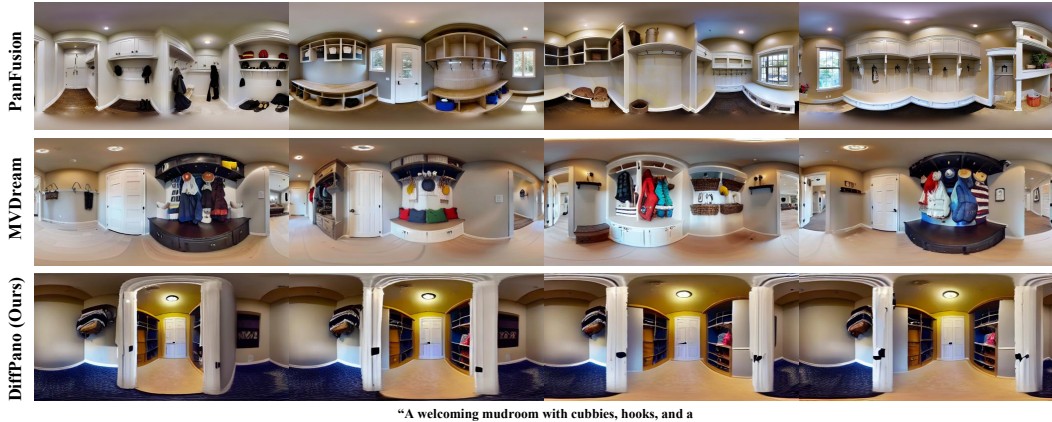

Figure 9: **Qualitative Comparisons of Text to Panoramic Videos.** Ours vs MVDream vs PanFusion.

multi-view panoramas generation, we conducted training on 8 80G A100 GPUs with a batch size of 1 and a learning rate of 1e-5. Each GPU utilized approximately 50% of its memory. The two-stage training process involved 100 epochs for each stage, with a total training time of approximately 5 days.

## D    Societal Impact

Since our method can achieve scalable, consistent, and diverse multi-view panoramas, it has many potential applications, such as unlimited room roaming in VR, interior design preview, embodied intelligent robot exploration, etc.

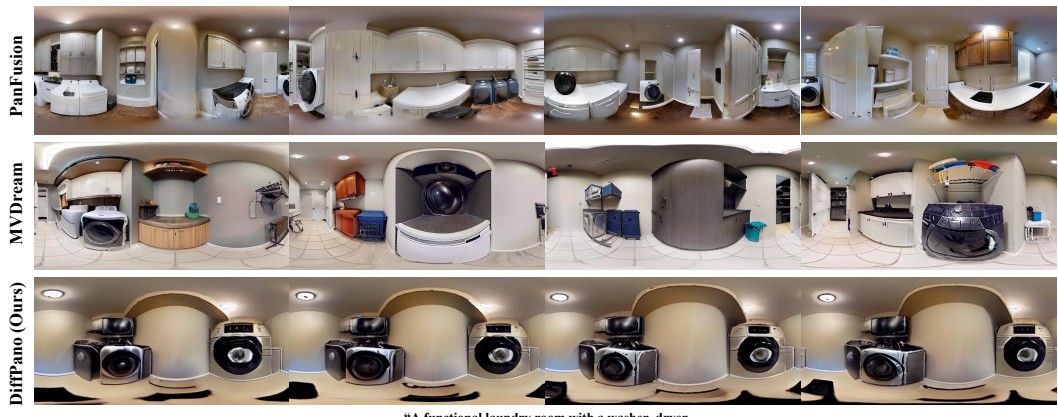

Figure 10: **Qualitative Comparisons of Text to Panoramic Videos.** Ours vs MVDream vs PanFusion.

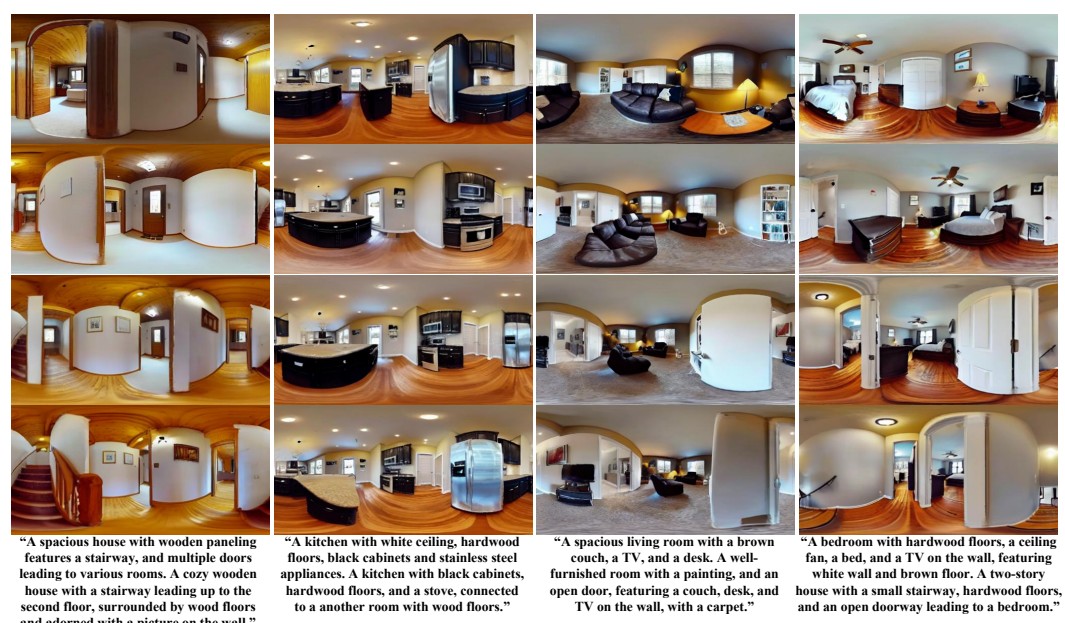

"A spacious house with wooden paneling features a stairway, and multiple doors leading to various rooms. A cozy wooden house with a stairway leading up to the second floor, surrounded by wood floors and adorned with a picture on the wall."

"A kitchen with white ceiling, hardwood floors, black cabinets and stainless steel appliances. A kitchen with black cabinets, hardwood floors, and a stove, connected to a another room with wood floors."

"A spacious living room with a brown couch, a TV, and a desk. A well-furnished room with a painting, and an open door, featuring a couch, desk, and TV on the wall, with a carpet."

"A bedroom with hardwood floors, a ceiling fan, a bed, and a TV on the wall, featuring white wall and brown floor. A two-story house with a small stairway, hardwood floors, and an open doorway leading to a bedroom."

Figure 11: **Qualitative Results of Text to Panoramic Videos.** DiffPano can generate scalable and consistent panorama videos.

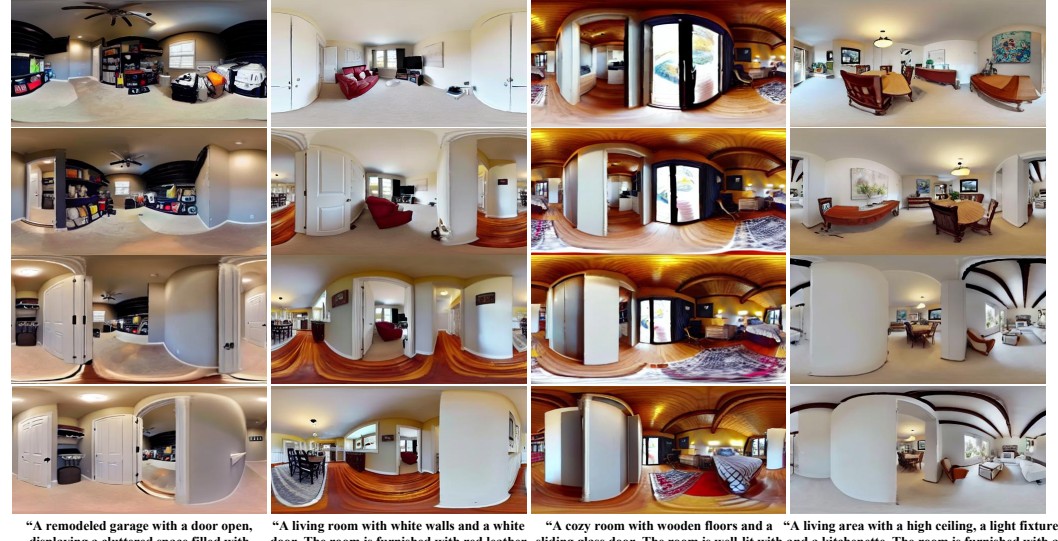

"A remodeled garage with a door open, displaying a cluttered space filled with various items on the walls. A well-lit hallway with a door and various furniture."

"A living room with white walls and a white door. The room is furnished with red leather couches and a TV mounted on the wall. A spacious room with hardwood floors, a kitchen and dining area, and an old-fashioned touch."

"A cozy room with wooden floors and a sliding glass door. The room is well-lit with a sink and wooden furniture. A cozy bedroom with wooden floors and a large bed. The room is furnished with a rug."

"A living area with a high ceiling, a light fixture, and a kitchenette. The room is furnished with a dining table and chairs and a painting on the wall, featuring a living room and dining room combination, and a white couch."

Figure 12: **Qualitative Results of Text to Panoramic Videos.** DiffPano can generate scalable and consistent panorama videos.

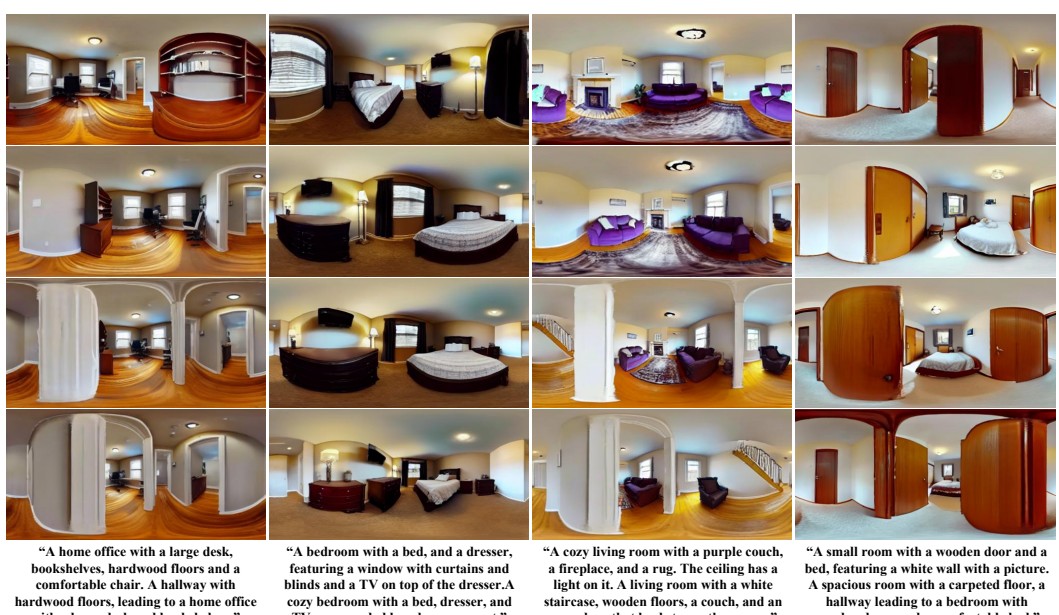

"A home office with a large desk, bookshelves, hardwood floors and a comfortable chair. A hallway with hardwood floors, leading to a home office with a large desk and bookshelves."

"A bedroom with a bed, and a dresser, featuring a window with curtains and blinds and a TV on top of the dresser.A cozy bedroom with a bed, dresser, and TV, surrounded by a brown carpet."

"A cozy living room with a purple couch, a fireplace, and a rug. The ceiling has a light on it. A living room with a white staircase, wooden floors, a couch, and an open door that leads to another room."

"A small room with a wooden door and a bed, featuring a white wall with a picture. A spacious room with a carpeted floor, a hallway leading to a bedroom with wooden doors and a comfortable bed."

Figure 13: **Qualitative Results of Text to Panoramic Videos.** DiffPano can generate scalable and consistent panorama videos.

Table 7: Network architecture of DiffPano-2

| | Layer | Output | Additional Inputs |
|---|---|---|---|
| | | CrossAttnUpBlock2 | |
| (37) | ResBlock | $640 \times 16 \times 32$ | (11), Time emb. |
| (38) | AttnBlock | $640 \times 16 \times 32$ | Prompt emb. |
| (39) | ResBlock | $640 \times 16 \times 32$ | (9), Time emb. |
| (40) | AttnBlock | $640 \times 16 \times 32$ | Prompt emb. |
| (41) | ResBlock | $640 \times 16 \times 32$ | (7), Time emb. |
| (42) | AttnBlock | $640 \times 16 \times 32$ | Prompt emb. |
| **(43)** | **EAModule** | **$640 \times 16 \times 32$** | |
| (44) | UpSampler | $640 \times 32 \times 64$ | |
| | | CrossAttnUpBlock3 | |
| (45) | ResBlock | $320 \times 32 \times 64$ | (6), Time emb. |
| (46) | AttnBlock | $320 \times 32 \times 64$ | Prompt emb. |
| (47) | ResBlock | $320 \times 32 \times 64$ | (4), Time emb. |
| (48) | AttnBlock | $320 \times 32 \times 64$ | Prompt emb. |
| (49) | ResBlock | $320 \times 32 \times 64$ | (2), Time emb. |
| (50) | AttnBlock | $320 \times 32 \times 64$ | Prompt emb. |
| **(51)** | **EAModule** | **$320 \times 32 \times 64$** | |
| (52) | GroupNorm | $320 \times 32 \times 64$ | |
| (53) | SiLU | $320 \times 32 \times 64$ | |
| (54) | Conv. | $4 \times 32 \times 64$ | |

