# OpenReview forum: "DiffPano: Scalable and Consistent Text to Panorama Generation with Spherical Epipolar-Aware Diffusion"
_NeurIPS.cc/2024/Conference — NeurIPS 2024 poster_

### Official Review · Reviewer_Yygs · 2024-07-09

**Soundness:** 2
**Presentation:** 2
**Contribution:** 2
**Rating:** 5
**Confidence:** 4

**Summary:**

This paper introduces the DiffPano framework, designed to generate consistent panoramic images from multiple viewpoints based on a text description of a scene. The authors first create a panoramic video-text dataset from 3D scenes using Habitat Simulator, BLIP2, and an LLM. Using this dataset, they fine-tune the original Stable Diffusion model with LoRA to develop a “single-view panorama-based Stable Diffusion.” This single-view model is then extended to a “multi-view consistent panorama Stable Diffusion” by incorporating a spherical epipolar attention module into the U-Net. Experiments show that DiffPano produces more consistent and higher-quality multi-view panoramic images compared to baseline methods.

**Strengths:**

1. The paper defines an interesting task of generating consistent panoramic images from multiple viewpoints, which could be useful for practical applications such as VR environments.
2. Providing a new dataset specifically designed for multi-view panoramic generation would be quite meaningful, facilitating further studies on panorama generation.
3. While epipolar-aware attention has been employed in prior works for perspective images [1], this paper effectively adapts a similar concept to panoramic images through a mathematical derivation for the spherical epipolar line.

[1] EpiDiff: Enhancing Multi-View Synthesis via Localized Epipolar-Constrained Diffusion, Huang et al., CVPR 2024

**Weaknesses:**

1. In Sec. 4.1, the authors describe panorama generation as a “style transformation” from the original perspective generation. However, generating a panoramic image requires more than a style change, as a 360-degree panorama must adhere to certain geometric constraints (i.e., when a portion of the panorama is projected into a perspective image, it should appear realistic). To address this, other than the user study, there need to be clear quantitative or qualitative evaluations of the realism of the generated multi-view panoramas. For instance, FAED [1] could be used to compare the realism of the panoramas themselves. Without such evidence, it is concerning whether PanoDiff is indeed generating more geometrically “realistic” panoramic images compared to the MVDream-based baselines for multi-view generation.
2. This work does not provide any qualitative examples of the perspective projections of the generated panoramas. In previous works on 360 panorama generation, the realism of the perspective view has been a crucial source for assessing the realism of the generated panoramas [2, 3]. Including perspective views would enhance the proper evaluation of the generation quality of the proposed method.
3. While the paper claims that DiffPano shows high generalizability, there are concerns regarding its performance on diverse text prompts. The single-view model is initially fine-tuned on panoramic images using LoRA, and the multi-view model is subsequently trained based on this fine-tuned model. This two-stage fine-tuning process on panoramic data might hinder the original generation diversity of Stable Diffusion, which is primarily trained on perspective images.

[1] Bips: Bi-modal indoor panorama synthesis via residual depth-aided adversarial learning, Oh et al., ECCV 2022

[2] MVDiffusion: Enabling Holistic Multi-view Image Generation with Correspondence-Aware Diffusion, Tang et al., NeurIPS 2023

[3] Taming Stable Diffusion for Text to 360 Panorama Image Generation, Zhang et al., CVPR 2024

**Questions:**

Missing citation: The data collection process for “Panorama Video Construction and Caption” (Fig. 2) appears to be quite related to the data captioning method proposed in Cap3D [1]. Can you elaborate on how it differs from or is related to their approach? If it is based on their approach, proper citation for the dataset generation pipeline is necessary.

[1] Scalable 3D Captioning with Pretrained Models, Luo et al., NeurIPS 2023

**Limitations:**

The paper discusses the limitation of the proposed method (hallucination of content).

---

> ### Author Rebuttal · Authors · 2024-08-07
>
> We thank Reviewer Yygs for his constructive suggestions and appreciate comments like "interesting task", "useful for practical applications such as VR environments", "new dataset quite meaningful", and "effectively adapts to panoramic images".
> We will gladly incorporate all the feedback in the revised version.
>
> 1) The main concern of reviewer Yygs is the quantitative or qualitative evaluations of the realism of the generated multi-view panoramas. We adopt the FAED metric to evaluate the realism. Experiments show that
> our method achieves lower FAED than MVDream and Text2Light, but slightly higher than PanFusion.
>
> | Methods | FID↓ | IS↑ | CS↑ | 	FAED↓ | Inference time↓ | LRMP↑ |
> |:-------:|:----:|:----:|:-------:|:-------:|:-------:|:-------:|
> | Text2Light |  76.5   |  3.60   | 27.48   | 97.24  | 80.6s | 127 |
> | PanFusion | 47.62 | 4.49 | 28.73 | 6.36 | 15.1s | 133 |
> | MVDream | 57.78 | 2.87 | 27.69 |  11. 63| 5.1s | 138 |
> | Ours	| 48.52| 	3.30| 	28.98| 	10.70| 5.1s| 149
>
> However, our CLIP Score achieves the highest results. Note that our method is significantly faster than existing methods, making scalable panoramas possible.
> The reason is that PanFusion adopts several complex modules in single-view panorama generation, which can improve FAED to a certain extent, but the speed is too slow. A fast pre-training module makes fine-tuning faster and better during multi-view generative network training.
>
> 2) Transform the panorama to the perspective views: As suggested, we converted the generated panorama into perspective images and conducted qualitative (shown in global response) and quantitative comparisons. Experiments show that our method achieves the lowest FID, while our method is higher than MVDiffusion in IS and slightly lower than PanFusion. It should be noted that MVDiffusion directly generates perspective images and then stitches them into panoramas. It is not in the ERP format and does not have the top and bottom parts. Our panorama generation speed is faster.
>
> | Methods | FID↓ | IS↑ |
> |:-------:|:----:|:----:|
> | MVDiffusion |  188.16   |  1.82   |
> | PanFusion | 213.19 |2.61|
> | Ours	| 164.75| 1.96|
>
> 3) Diversity of DiffPano: We show some qualitative panorama results in the global response, which shows our method has good diversity. We also show some outdoor panoramic generation results to demonstrate generalization. Generalization and panorama effects are a trade-off. When we pursue better geometric consistency in panorama generation, generalization may indeed decrease.
> This requires a balance. A potential solution is to use more panoramic images to train native text-to-panorama generation diffusion models, rather than using LoRA to fine-tune from SD, which can be a future work. However, training such a model from scratch requires lots of GPU and dataset. Using LoRA to fine-tune the SD model may be a more practical and cost-effective approach.
>
> 4) One-stage vs Two-stage:
> We conducted ablation experiments on one-stage and two-stage training. Experiments show that due to the introduction of the epi module, the multi-view consistency of the two training methods is relatively good. Comparing the consistency is not enough to measure the difference between the two methods. The IS of the two-stage training method is lower, and the diversity is reduced to a certain extent, which is slightly worse than the one-stage training. However, it should be noted that the images after one-stage training will have ghosting, but the two-stage will not. We show the qualitative comparison results in the global response (the FID value of the two-stage method will also be higher).
>
>
> | Methods | IS↑ | FID↓	| LPIPS↓| 	PSNR↑ | SSIM↑|
> |:-------:|:----:|:----:|:-------:|:-------:|:-------:|
> | One-stage |  4.52   |  82.92   | 0.0454   | 31.64  | 0.74 |
> | Two-stage | 3.13 | 74.08 | 0.0610 | 31.54 | 0.73 |
>
> 5) Missing citation of Cap3D: First of all, the objects of text annotation are different. Cap3D annotates objects, while we annotate scenes. We first annotate the perspective view of the cube map. When integrating different perspective views, the instructions input to LLM are specially processed for the scene to prevent redundancy and other problems. We show the difference in text description before and after LLM integration in Figure 2 of this article. Our LLM text prompt is as follows:
>
> You will be given six text prompts that describe the appearance of a room. Your task is to summarize these six given text prompts into a single text prompt that describes what the room looks like. This single summary text prompt in your response should strictly begin with <begin> and end with </end>. Simplify the room types if there are multiple room types in the prompts. The final text prompt should be less than 77 tokens and do not omit any furniture or any appearance details in the given prompts.
>
> As suggested, we will cite Cap3D in the revised version.

---

> > ### Comment · Reviewer_Yygs · 2024-08-13
> >
> > Thank you to the authors for providing a detailed rebuttal on the raised concerns. I have carefully read the rebuttal and would like to add a few comments.
> >
> > Firstly, considering the new qualitative results on perspective views and diverse generation outputs, I acknowledge that DiffPano indeed demonstrates the capability in generating panoramas that are realistic to a certain extent. Additionally, the additional experiment for FAED suggests that DiffPano outperforms its MVDream baseline. **In light of this, I raise my evaluation to "Borderline Accept."**
> >
> > **However, there are still a few aspects that remain unclear. If the authors could provide some comments on the below issues, it would greatly assist in the following discussion between the reviewers and ACs:**
> >
> > * I am not entirely convinced that DiffPano's "highest CLIP score" can support the claim of generating "realistic" panoramas. While CLIP score measures text-image alignment, it is not clear how it directly correlates with the realism of the generated panoramas.
> >
> > * The statement that *"making scalable panoramas possible"* remains ambiguous. Given its importance in justifying the lower scores of DiffPano compared to PanFusion, it would be helpful if you could elaborate on this point. I still question whether achieving fast inference is worth the significant trade-off in generation quality/realism (as seen in the notable difference between PanFusion and DiffPano on the FAED metric).
> >
> > Thank you again for the additional information and for your continued efforts in addressing these concerns.

---

> > > ### Author Response · Authors · 2024-08-13
> > > **Reply to Reviewer Yygs and please let us know if your concerns have been addressed**
> > >
> > > Dear Reviewer Yygs,
> > >
> > > Thanks for your reply. **Fr´echet Inception Distance (FID) and Fr´echet Auto-Encoder Distance (FAED) are used to compare realism.** FID is widely used for image generation, it relies on an Inception network trained on perspective images, thus less applicable for panoramic images. Therefore, a variant of FID customized for panorama, **FAED is used to better compare the realism.** **Inception Score (IS) is used to measure realism the diversity.** **CLIP Score (CS) is used to evaluate the text-image consistency.** From the rebuttal, **our CLIP Score achieves the highest results, which means that our method can generate panoramas that are more consistent with the text description.** Although the panorama generated by PanFusion can obtain slightly higher FID and FAED than our method, we found that **when the panorama is converted to perspective, we can obtain higher FID, which shows that our perspective has better realism**. It should be noted that **PanFusion is only a single-view panorama generation. It improves the realism of the panorama by adding many complex modules but reduces the realism of the perspective view converted from panorama**. In addition, **the top and bottom of the panorama generated by PanFusion is blurred**, and **the speed of its panorama generation is too low, only about 1/3 of the speed of our single panorama generation method**. **We can achieve a relatively realistic panorama that is more consistent with the text description through style transfer and simple data augmentation, and the speed is greatly improved**. **A faster pre-training module makes fine-tuning faster and better during multi-view generative network training.**  **The core of this paper is to perform scalable multi-view panorama generation**. With the faster generation speed, we can **generate longer panoramic videos in a shorter time, which is beneficial to subsequent downstream applications.**
> > >
> > > Why does fast inference make scalable panoramas possible? As mentioned above, although PanFusion can obtain a slightly higher FAED score for panoramas, it should be noted that **the top and bottom of the panorama generated by PanFusion is blurred**, and **the speed of its panorama generation is too low**. While our single-view panorama generation can generate a relatively realistic panorama that is more consistent with the text description. The speed is about 3x of PanFusion's. **Our generated panoramas have a clear top and bottom**. Since **this paper focuses on multi-view panorama generation, PanFusion can not keep the consistency of multi-view panorama generation, we are the first method that can generate consistent and scalable multi-view panoramas**. To achieve this, we **propose spherical epipolar multi-view diffusion fine-tuned on the single-view panorama generation method. A faster pre-training module makes the fine-tuning of multi-view diffusion faster and better during the training of the multi-view generative network**. **The core of this paper is to perform scalable multi-view panorama generation**. With the faster generation speed, we can **generate longer panoramic videos in a shorter time, which is beneficial to subsequent downstream applications, such as VR roaming.**
> > >
> > > **We hope that our rebuttal addresses your questions and concerns**. As the discussion phase is nearing its end, **we would be grateful to hear your feedback and wondered if you might still have any concerns we could address.**
> > >
> > > **It would be appreciated if you could raise your score on our paper**. We thank you again for your effort in reviewing our paper.
> > >
> > > Best regards,
> > >
> > > DiffPano Authors

---

> ### Author Response · Authors · 2024-08-13
> **Please let us know if your concerns have been addressed**
>
> Dear Reviewer Yygs,
>
> Thank you again for your review. **We hope that our rebuttal could address your questions and concerns** such as the quantitative or qualitative evaluations of the realism of the generated multi-view panoramas, the qualitative examples of the perspective projections of the generated panoramas, the generalizability of DiffPano, and the citation of Cap3D. As the discussion phase is nearing its end, **we would be grateful to hear your feedback and wondered if you might still have any concerns we could address**.
>
> We have **provided an anonymous link to the AC to forward the link to the reviewers, due to the requirements during the rebuttal stage.** This anonymous link contains some generated outdoor panoramic samples.
>
> **It would be appreciated if you could raise your score on our paper**.
> We thank you again for your effort in reviewing our paper.
>
> Best regards,
>
> DiffPano Authors

---

### Official Review · Reviewer_K8Nd · 2024-07-11

**Soundness:** 2
**Presentation:** 3
**Contribution:** 2
**Rating:** 5
**Confidence:** 5

**Summary:**

The paper focusing on the simulation of panoramic images and the annotation of text descriptions. The primary contribution is the introduction of a LoRA-based fine-tuning technique, aimed at enhancing the performance of benchmark datasets. The authors propose a pipeline that integrates simulated panoramic data with annotated textual descriptions to improve the accuracy and robustness of 3D models.

**Strengths:**

The idea of simulating panoramic images combined with text annotation is interesting, potentially offering new ways to enhance 3D vision using panorama as representation.
The paper is well-structured, with a clear explanation of the proposed method and the experimental setup.

**Weaknesses:**

Lack of Technical Contribution: The paper primarily focuses on the application of existing techniques (panoramic simulation, text annotation and LoRA-based fine-tuning) rather than introducing new technical innovations. This limits its contribution to the field.

Typos and Clarity Issues: There are several typos and unclear sentences throughout the paper (e.g., "??" in line 34), which detract from its overall readability and professionalism.

The scope of this paper lies in the indoor scene panorama generation, and it is suggested to highlight this keyword in both title and abstract.

**Questions:**

Can the authors clarify the experimental setup and parameters used for LoRA-based fine-tuning? More details on the hyperparameters and training process would be helpful.

Are there any specific challenges or limitations encountered when simulating panoramic images (especially the difficulty for outdoor/synthetic simulation)? How were these addressed in the proposed pipeline?

**Limitations:**

Although this paper introduce an interesting dataset and problem, it is unclear how was the dataset quality, as the supplementary materials are empty. Could the author clarify the date for releasing the datasets? The scope of the paper is more close to the benchmark and dataset track where the datasets can be used to insight the 3D computer vision researches.
Besides, the authors should discuss the scalability of their approach to larger and more complex scenarios (indoor, outdoor, synthetic, cartoon). Additionally, they should address any potential biases introduced by the simulated data and how these might affect the generalizability of the results.

---

> ### Author Rebuttal · Authors · 2024-08-07
>
> We thank Reviewer K8Nd for his constructive suggestions and appreciate comments like "interesting idea", "new way to enhance 3D Vision", "well-structured", and "clear explanation". However, the reviewer seems to have misunderstood the core content of this paper,
> the problem it solves, and the core innovation. Here, we point out the factual error.
>
> This paper aims to perform scalable multi-view panorama generation with text (which can be expanded from one room to another). As far as we know, we are the first to propose text-to-multi-view panorama generation. Previous methods are limited to single-view panorama generation. The core reason is the lack of large-scale multi-view panorama datasets, and single-view panorama generation cannot guarantee the multi-view consistency of the generated panorama. A new framework suitable for multi-view panorama generation is needed.
> To this end, we first construct a large-scale panoramic video-text dataset. Then, we design a multi-view panorama generation framework. First, we propose a diffusion model for text-to-single-view panorama generation (fine-tuned on the newly proposed panorama dataset using LoRA). To achieve multi-view consistency, we derive a spherical epipolar constraint suitable for panoramas and propose the spherical epipolar multi-view panorama generation framework, which can achieve diverse and consistent scalable panorama generation results. We will gladly incorporate all the feedback in the revised version.
>
> 1) Summary: As mentioned above, the reviewer seems to have misunderstood the core mission, problem, solution, and core innovation of this paper. Other reviewers (2LuU, 7HQE, Yygs) can understand the content of this paper relatively well.
> We suggest that the reviewer re-review this paper to make a reasonable evaluation.
> We are glad to resolve any doubts of the reviewer. This paper first introduces a novel task for text to multi-view panorama generation.
> So we need to reconstruct a large-scale panoramic video-text dataset for this task and design a novel multi-view panorama generation framework with novel spherical epipolar attention. This method can generate diverse and consistent scalable panorama generation.
>
> 2) Lack of Technical Contribution: The paper focuses on text-to-multi-view panorama generation, rather than single-view panorama generation. It is the essential difference between this paper and the existing single-view panorama generation method.
> We propose a novel panorama video construction and caption pipeline and create a large-scale panorama video-text dataset.
> Then we design a novel multi-view panorama generation framework with novel spherical epipolar attention.
>
> 3) Indoor scene panorama generation: Although this paper focuses on the multi-view panoramic generation of indoor scenes,
> we also show the results of some outdoor scene generation of this method in the global response, which shows a certain generalization.
> In the future, we will explore outdoor scene reconstruction to construct large-scale outdoor panoramic video datasets
> to further improve the generalization of multi-view panoramic generation.
>
> 4) LoRA-based fine-tuning: It should be emphasized that Lora-based fine-tuning is a method for text to single-view panorama generation
> (which is detailed introduced in Section 4.1), and this paper focuses on text-to-multi-view panorama generation
> (which is detailed introduced in Sec. 4 and 5 and Appendix C).
> Here we add the details of Lora-based fine-tuning: We first fix the stable diffusion model 1.5 and use Lora for fine-tuning.
> The fine-tuning dataset is selected from our newly proposed video panoramic-text dataset to form 8508 high-quality text-panorama pairs.
> The learning rate is 0.0001, the batch size is 32, and 8 A100 80GB are used to train 100 epochs.
>
> 5) Any specific challenges or limitations encountered when simulating panoramic images?
> Our synthetic panoramic video-text dataset is based on Matterport3D, which is reconstructed from the captured real images.
> Based on this dataset, there will be no domain gap problem at least when generating indoor panoramas.
> We also try to perform outdoor panora generation, which is shown in the global response.
> Experiments show that our method has a certain degree of generalization in outdoor panorama generation.
> Of course, if we can add a more high-quality outdoor dataset, it may make our model more generalizable. We leave it as future work.
>
> 6) Dataset quality of the proposed panoramic video-text dataset? releasing date?
> In the global response, we show the comparison between our dataset and PanFusion dataset.
> Compared with the PanFusion dataset, our dataset has a clear top and bottom, and the text description is more detailed.
> Since the complete dataset is very large, we plan to release the train/test dataset and
> open source a complete panoramic video construction and caption pipelines upon acceptance,
> so that the community can create large-scale panoramic video-text datasets according to their needs.
> We will also release the complete framework upon acceptance so that the community can use and follow our work.
>
>
> 7) Closer to benchmark and dataset track? No, as mentioned above, we propose a novel task of text to multi-view panorama generation and propose novel panoramic video construction and caption pipelines to create a large-scale panoramic video-text dataset. Then we design a novel panorama generation framework, consisting of a single-view panorama generation module and a novel spherical epipolar multi-view panorama generation module.
> It's not just a dataset; the dataset is just one of our core contributions. This paper is suitable for the main paper track.
>
> 8) Scalability and generalizability: We demonstrate panoramic video generation and outdoor panorama generation, which shows that our method has good scalability and generalization ability.
>
> 9) Typos: We will polish the writing and proofread the revised version, including the typos.

---

> > ### Comment · Reviewer_K8Nd · 2024-08-12
> > **Replies**
> >
> > Thank you to the authors for providing comprehensive responses to clarify the work.
> >
> > The proposed work clearly remains within the scope of **text-to-panoramic images generation**, creating a large-scale dataset using multi-view panoramic images for each sample. The application of LoRA-based fine-tuning combined with spherical epipolar attention to address new generation problem.
> >
> > After reviewing the clarifications and the visualized PDF, some of the concerns regarding the contribution formulation—specifically whether the methodology/task or the panorama generation pipeline is being overemphasized, and the generalization for outdoor cases—have been partially addressed. However, there are still issues that need attention, such as the generated realism, the lack of qualitative examples, and the discussion on generalization to avoid overclaiming in the paper. These should be thoroughly discussed in the revised submission:
> >
> > - High-resolution (original) data samples should be provided in the supplementary materials to allow for detailed inspection.
> > - Why does Figure 6 in the attached PDF contain significant artifacts at the boundaries? Are these artifacts caused by the low resolution of the generated images, or is it due to the method treating the generation process as a "style transformation," potentially lacking geometric constraints?
> > - There is a small typo in the attached Figure 5 where the closing bracket is missing.
> >
> > If the issues mentioned above can be further addressed, I would be willing to raise my recommendation to borderline accept based on the current quality of the submission, the visualizations provided in the response, and feedbacks from other reviewers. Addressing these new concerns in depth would greatly benefit the discussion within the reviewer-AC panel.
> >
> > Thanks.

---

> > > ### Author Response · Authors · 2024-08-13
> > > **reply to the reviewer K8Nd**
> > >
> > > Dear Reviewer K8Nd，
> > >
> > > Thank you for your reply and for expressing the recommendation to borderline accept our paper. We are glad to see that our comprehensive responses can **correct your misunderstanding to some extent**. However, we still want to clarify that this paper does belong to the field of text-to-panoramic image generation. However, unlike **existing methods focusing on single-view panorama generation**, **this paper focuses on scalable multi-view panorama generation**. **This paper is the first work on text-to-multi-view panorama generation.** To this end, we **propose a novel panoramic video-text pipeline to build a large-scale panoramic video-text dataset.** To generate multi-view panoramas, we **propose a spherical epipolar multi-view diffusion model** for the multi-view panorama generation, which **improves the multi-view consistency of the generated panoramas** by exploiting the geometric properties of the spherical epipolar of the panoramas. We **treat a single-view panorama generation as style transfer and use simple data augmentation to improve the geometric properties of the left-right consistency of the panorama, making scalable panorama generation possible**. It should be noted that **our network is compact, faster (~3x speed) than single-view panorama generation method like PanFusion, and easier to extend to multi-view panoramas**.
> > >
> > > As for the generated realism, as the rebuttal mentioned, our synthetic panoramic video-text dataset is based on Matterport3D, which is reconstructed from the captured real images. There will be **no domain gap problem at least when generating indoor panoramas**. As for generalization, we perform outdoor panorama generation, which shows **our method has a certain degree of generalization in outdoor panorama generation, we will add this in the revised version.** Of course, we will **explore high-fidelity outdoor reconstruction for creating more diverse panoramic video-text datasets, which may make our model more generalizable. We leave it as future work.**  As for qualitative examples,  we **showcase text-to-multi-view panorama generation in the supplementary material**, and **provide an anonymous link to the AC to forward the link to the reviewers**, due to the requirements during the rebuttal stage.
> > >
> > > We will **showcase more high-resolution panoramic video datasets with detailed text descriptions in the supplementary materials of the revised version**. We also plan to **release the train/test dataset and open source a complete panoramic video construction and caption pipelines and release the complete framework** upon acceptance, so the community can create large-scale panoramic video-text datasets according to their needs and use and follow our work.
> > >
> > > Since our method is to directly generate panoramas with text description, **directly converting the panorama into perspective views may face a certain precision loss due to numerical interpolation**. In addition, **the resolution of the currently generated panorama is relatively low, and directly converting it to a perspective view will also cause artefacts**. **Training high-resolution panorama generation requires more resources and more time**. We can later **explore the upsampling method in video generation to fine-tune higher-resolution panorama generation to reduce artefacts**. Although our method regards the single-view generation as a style transfer from perspective view to panorama, since we added data augmentation during the training process to improve the geometric constraints of the left and right consistency of the panorama, this impact is relatively small.
> > >
> > > We will polish the writing and proofread the revised version and the supplementary material, including the typos.
> > >
> > > We hope that our response addresses your concerns. **It would be appreciated if you could further raise your score on our paper.** We thank you again for your effort in reviewing our paper.
> > >
> > > Best regards,
> > >
> > > DiffPano Authors

---

> ### Author Response · Authors · 2024-08-07
> **To SAC for pointing out the factual error of Reviewer K8Nd**
>
> We thank all reviewers for their constructive comments and recognition of our work ("first work, pioneer work, interesting task", "novel and technically sound", "plausible and visually pleasant panoramic results"), and ACs and SACs for their hard work.
> However, Reviewer K8Nd seems to have misunderstood the core content of this paper,
> the problem it solves, and the core innovation. We believe this review may be pure LLM generated.
> We hope that ACs, SACs, and all reviewers can read our paper and the rebuttal and then have in-depth discussions on K8Nd's misunderstandings to ensure that reviewer K8Nd can eliminate the factual errors and re-evaluate our work to obtain a fair review.
> We sincerely thank you.
>
> Here, we point out the factual error. Reviewer K8Nd argues that our approach lacks innovation and is a technique that has been studied in previous approaches. The goal of this paper is to perform scalable multi-view panorama generation based on text (which can be expanded from one room to another). As far as we know, we are the first to propose text-to-multi-view panorama generation. Previous methods are limited to single-image panorama generation. The core reason is the lack of large-scale multi-view panorama datasets, and single-view panorama generation work cannot guarantee the multi-view consistency of the generated panorama. A new framework suitable for multi-view panorama generation is needed. To this end, we first construct a large-scale panoramic video-text dataset. Then, we design a multi-view panorama generation framework. First, we propose a diffusion model for text-to-single-view panorama generation (fine-tuned on the newly proposed panorama dataset using LoRA). To achieve multi-view consistency, we derive a spherical epipolar constraint suitable for panoramas and embed it into the network as an attention layer to realize the spherical epipolar multi-view panorama generation framework, which can achieve diverse and consistent scalable panorama generation results. Our method is different from previous methods in terms of both technique and dataset (as recognized by Reviewer 2LuU and Reviewer 7HQE), but Reviewer K8Nd did not realize the innovativeness of our technique.
>
> We hope that our rebuttal and explanation can correct the reviewer K8Nd's misunderstanding
> so that our work can be properly reviewed and obtain a satisfactory review result.
> We sincerely hope that ACs and SACs can have in-depth discussions with all reviewers to review our work more fairly.

---

> ### Author Response · Authors · 2024-08-13
> **Please let us know if your concerns have been addressed**
>
> Dear Reviewer K8Nd,
>
> **We hope that our rebuttal addresses your questions and concerns.** As the discussion phase is nearing its end, **we would be grateful to hear your feedback and wondered if you might still have any concerns we could address.**
>
> **It would be appreciated if you could raise your score on our paper if we address your concerns**. We thank you again for your effort in reviewing our paper.
>
> Best regards,
>
> DiffPano Authors

---

### Official Review · Reviewer_7HQE · 2024-07-12

**Soundness:** 3
**Presentation:** 3
**Contribution:** 3
**Rating:** 6
**Confidence:** 5

**Summary:**

This work proposes a text-driven panorama generation framework to achieve scalable, consistent, and diverse panoramic scene generation. In particular, a spherical epipolar attention module with relative poses is designed to ensure multi-view consistency. Moreover, a comprehensive panoramic video-text dataset is constructed, which contains millions of consecutive panoramic frames with corresponding depths, camera poses, and text descriptions. Extensive experiments demonstrate the superiority of the proposed method beyond previous works.

**Strengths:**

+ This work is a pioneer in exploring the scalable multi-view panorama generation task from text descriptions.
+ A diverse and rich panoramic video-text dataset, which shows promising potential to promote the community of panoramic generation.
+ The proposed framework looks compact and practical to implement.
+ Overall, I am in favor of the presentation of this work, it is well-structured and easy to follow. The generated panoramic results also look plausible and visually pleasant.

**Weaknesses:**

- For the motivation of constructing the panoramic video-text dataset, the authors argue that the corresponding text descriptions in previous panoramic datasets are not precise enough. However, such a conclusion needs to be supported by some quantitative metrics or data statistics.
- The seamless content is an important property in the generated panoramas. Namely, the panorama can be stitched from left to right boundaries without a noticeable edge effect. This work performs data augmentation to improve such a left-right continuity. Have the authors considered the cylinder convolutions like previous panoramic vision works? For example, "Cylin-Painting: Seamless 360° Panoramic Image Outpainting and Beyond" and "Spherical Image Generation From a Few Normal-Field-of-View Images by Considering Scene Symmetry" use a cylinder-like convolution or circular padding to ensure the seamless content generation of panoramas. The brief reviews and discussions of these works are also expected to be presented in this work.
- Some previous works propose specific metrics to evaluate the panorama model. For example, "PanoFormer: Panorama Transformer for Indoor 360 Depth Estimation" proposes the Left-Right Consistency Error (LRCE) to quantitatively measure the consistency of the left-right boundaries by calculating the horizontal gradient. The authors are suggested to add this metric for generated panorama evaluations in experiments.
- Minor points: There are some typos in the manuscript. For example, in line 34: "(see Sec. ??)".

**Questions:**

Can the proposed framework be extended to panoramic video generation? More insightful discussions are expected to be provided.

**Limitations:**

This work lacks some customized metrics for evaluating the property of panoramic images, which should be different from those designed for the perspective images.

---

> ### Author Rebuttal · Authors · 2024-08-07
>
> We thank reviewer 7HQE for his constructive suggestions and appreciate comments like “pioneer work in scalable multi-view panorama generation", "diverse and rich panoramic video-text dataset promote the community of panoramic generation", "compact and practical framework", "well-structured and easy to follow", "plausible and visually pleasant panoramic results".
> We will gladly incorporate all the feedback in the revised version.
>
> 1) Text descriptions between previous panoramic datasets (PanFusion) with our proposed panoramic video-text dataset:
> PanFusion uses BLIP2 to directly generate text descriptions for panoramas. It only has four or five words and is very concise.
> The CLIP Score (CS) cannot reflect the accuracy of the text description, and the PanFusion dataset has the problem of blurry top and bottom. In contrast, our panoramic video dataset construction pipeline first generates text descriptions for perspective images using BLIP2, then uses LLM to summarize, which can obtain more detailed text descriptions. At the same time, the top and bottom of our panoramas are clear, and the dataset is larger (millions of panoramic keyframes ). We also provide the camera pose of each panorama, the corresponding panoramic depth map, etc. We show the difference between the PanFusion dataset with our dataset in the global response.
>
> 2) Consider cylinder convolutions or padding to maintain left-right continuity:
> As suggested, we compare cylinder padding without data augmentation and the data augmentation method used in this paper to improve the left-right consistency. Experiments of 1092 samples show that our simple data augmentation scheme can obtain higher CLIP Score (CS) and Inception Score (IS), as well as lower Fréchet Inception Distance (FID). To further verify the consistency, we propose a new left-right consistency evaluation method. First, we simply cut the generated panorama into left and right images, and then use the image-matching algorithm to calculate the correspondence. The number of correspondences is called as Left-Right Matching Pairs (LRMP) metric. Experiments show that our method can obtain more correspondence.
> | Methods | IS↑ | FID↓ | CS↑ | LRMP↑ |
> |:-------:|:----:|:----:|:-------:|:-------:|
> | Circular Padding    |   3.81   |   68.31  | 23.51 | 143   |
> | Data Augmentation   |   4.07   |   61.43   | 23.58   | 149   |
>
> 3) Specific metrics for panorama, such as Left-Right Consistency Error (LRCE):
> The Left-Right Consistency Error (LRCE) metric proposed by PanoFormer evaluates the accuracy of panoramic depth estimation
> by comparing the gap between the leftmost and rightmost horizontal gradients of the predicted panoramic depth map
> and the ground-truth panoramic map. In the panorama generation task of this paper, we do not have the ground-truth panoramic depth,
> and this method does not generate panoramic depth. The LRCE metric is not applicable to the work of this paper.
> Instead, we propose a Left-Right Matching Pairs (LRMP) metric to calculate the left-right consistency by using matching algorithms.
> We follow Text2Light to report Fr´echet´s Inception Distance (FID) and Inception Score (IS) on panoramas to measure realism and diversity. Additionally, the CLIP Score (CS) is used to evaluate the text-image consistency. While FID is widely used for image
> generation, it relies on an Inception network trained on perspective images, thus less applicable for panorama images.
> We adopt a variant of FID customized for panorama, Fr´echet Auto-Encoder Distance (FAED) in PanFusion, which is used to better compare realism. The quantitative results are as follows:
> | Methods | FID↓ | IS↑ | CS↑ | 	FAED↓ | Inference time↓ | LRMP↑ |
> |:-------:|:----:|:----:|:-------:|:-------:|:-------:|:-------:|
> | Text2Light |  76.5   |  3.60   | 27.48   | 97.24  | 80.6s | 127 |
> | PanFusion | 47.62 | 4.49 | 28.73 | 6.36 | 15.1s | 133 |
> | Ours	| 48.52| 	3.30| 	28.98| 	10.70| 5.1s| 149
>
> Experiments show that our method achieves the best CLIP Score, the best left-right consistency, and the fastest inference speed, which makes scalable panorama generation possible. Although our method is slightly weaker than PanFusion in FID, IS, and FAED metrics, it cannot obtain clear images due to the blur at the top and bottom, and the complex network makes its inference speed too slow.
>
> 4) Extended to panoramic video generation? Yes, our method can be extended to panoramic video generation.
> This paper mainly studies the method of text-to-multi-view panorama generation.
> We have recently studied the method of image-to-multi-view panorama generation.
> To generate panoramic videos, we can first generate multi-view images of different rooms with large pose changes conditioned on the text, and then run the image-to-multi-view panorama generation conditioned on the generated multi-view panoramas,
> so that it can be expanded to generate longer panoramic videos. We will add this part to the revised version.
> We show some panoramic video effects in the pdf of the global response. It is recommended to open it with Adobe reader software so that you can see these videos.
>
> 5) typos: We will polish the writing and proofread the revised version, including the typos.

---

> ### Author Response · Authors · 2024-08-13
> **Please let us know if your concerns have been addressed**
>
> Dear Reviewer 7HQE,
>
> Thank you again for your review. **We hope that our rebuttal addresses your questions and concerns**, such as the motivation for constructing the panoramic video-text dataset, circular padding vs data augmentation, and specific metrics to evaluate the panorama model and panoramic video generation. As the discussion phase is nearing its end, **we would be grateful to hear your feedback and wondered if you might still have any concerns we could address.**
>
> We have **provided an anonymous link to the AC to forward the link to the reviewers**, due to the requirements during the rebuttal stage. This anonymous link contains **the difference between PanFusion's dataset and our proposed dataset**, **panoramic video generation**, etc.
>
> **It would be appreciated if you could raise your score on our paper if we address your concerns**. We thank you again for your effort in reviewing our paper.
>
> Best regards,
>
> DiffPano Authors

---

> > ### Comment · Reviewer_7HQE · 2024-08-13
> >
> > Thanks for the detailed rebuttal. I would like to increase the rating. The updated experimental results and discussions of the related works are expected to be presented in the final version.

---

> > > ### Author Response · Authors · 2024-08-13
> > > **Reply to Reviewer 7HQE**
> > >
> > > Dear Reviewer 7HQE,
> > >
> > > Thanks for increasing the rating and for your review. **We will add the updated experiment results and discussions of the related works in the revised version.**
> > >
> > > **It seems that our rebuttal addresses your questions and concerns.** As the discussion phase is nearing its end, **we would be grateful to hear your feedback and wondered if you might still have any concerns we could address**.
> > >
> > > **It would be appreciated if you could recommend accepting our paper in the reviewers-pc discussion**. We thank you again for your effort in reviewing our paper.
> > >
> > > Best regards,
> > >
> > > DiffPano Authors

---

### Official Review · Reviewer_2LuU · 2024-07-15

**Soundness:** 4
**Presentation:** 4
**Contribution:** 3
**Rating:** 6
**Confidence:** 3

**Summary:**

The paper presents a novel framework called DiffPano for scalable and consistent text-to-panorama generation. The authors first build a panoramic video-text dataset, then propose a spherical epipolar-aware diffusion model to generate multi-view consistent panoramic images, addressing the limitations of existing single-view panorama generation methods. Extensive experiments demonstrate that the proposed DiffPano framework can generate scalable, consistent, and diverse panoramic images from text descriptions.

**Strengths:**

- This paper is well-written and easy to follow.
- This is the first work that tackles the research direction of text to multi-view panorama generation.
- The authors have created a large-scale panoramic video-text dataset, which is a valuable resource for the community.
- The proposed spherical epipolar-aware diffusion model is a novel and technically sound approach to address the challenge of generating multi-view consistent panoramic images from text.
- The experimental results show the effectiveness of the DiffPano framework in generating scalable, consistent, and diverse panoramic images.

**Weaknesses:**

- It seems like this work can only do in-distribution generation (indoor scene generation only) and cannot generalize to arbitrary unbounded outdoor scenes.

- This paper does not address why generating multi-view ERP panoramas is meaningful. Is the multi-view output format more helpful for later 3D reconstruction? And why?

- Typos:

L34: see Sec. ??

**Questions:**

What is the use case for this work?

What kinds of downstream applications need multi-view panorama images?

---

> ### Author Rebuttal · Authors · 2024-08-07
>
> We thank the reviewer 2LuU for his constructive suggestions and
> appreciate comments like "well-written", "easy to follow",
> "first work for text to mv panorama", "valuable panoramic video-text dataset",
> "novel and technically sound approach of spherical epipolar-aware diffusion",
> "effectiveness in generating scalable, consistent, and diverse panoramic images".
> We will gladly incorporate all the feedback in the revised version.
>
> 1) The main concern of reviewer 2LuU is the generalizability of DiffPano and whether it can generate outdoor scenes.
> To our knowledge, there is no outdoor multi-view panoramic video dataset. Although DiffPano is currently only trained on the indoor multi-view panoramic dataset, as suggested by the reviewer, we try to generate outdoor panoramic scenes.
> In the global response, we generated some outdoor multi-view panoramas of DiffPano, which show that DiffPano has good generalization. Of course, it is not perfect. We can explore the 3D reconstruction of outdoor scenes and put the 3D scenes
> into the panoramic video-text data pipeline to create a large-scale outdoor panorama dataset to improve the generalization of DiffPano further. We leave this as future work.
>
> 2) Reviewer 2LuU is also concerned about “why generating multi-view ERP panoramas is meaningful”.
> A single-view panorama can only represent local areas, missing many parts of the scene.
> A multi-view panorama is controllably generated with camera poses, which can generate the missing parts of a single view.
> We visualized the difference between multi-view and single-view in the global response.
>
> 3) As for the use cases of this work, we support single-view panorama generation or multi-view panorama generation
> with text prompts and camera poses. We additionally provide a generation framework with image conditions,
> which can further achieve the effect of panoramic video generation, as shown in the global response. We will add it to the revised version.
>
> 4) Reviewer 2LuU would like to know about the downstream applications of multi-view panoramas.
> Multi-view panoramas have many applications, such as room roaming in VR (reviewer Yygs recognized this application),
> interior design preview, embodied intelligent robot exploration, etc. For example, in VR applications,
> existing single panorama generation can only obtain local information about scenes, leading to limited roaming scapes,
> while multi-view panorama generation can achieve unlimited roaming from one room to another.
>
> 5) Typos: We will polish the writing and proofread the revised version, including the typos.

---

> ### Author Response · Authors · 2024-08-13
> **Please let us know if your concerns have been addressed**
>
> Dear Reviewer 2LuU,
>
> Thank you again for your review. **We hope that our rebuttal could address your questions and concerns**, such as the generalizability of DiffPano, why generating multi-view ERP panoramas is meaningful, and the downstream applications of multi-view panorama images. As the discussion phase is nearing its end, **we would be grateful to hear your feedback and wondered if you might still have any concerns we could address.**
>
> We have **provided an anonymous link to the AC to forward the link to the reviewers, due to the requirements during the rebuttal stage**. This anonymous link which contains **outdoor panorama generation and single-view vs multi-view panoramas**.
>
> **It would be appreciated if you could raise your score on our paper if we address your concerns**.  We thank you again for your effort in reviewing our paper.
>
> Best regards,
>
> DiffPano Authors

---

> ### Author Response · Authors · 2024-08-13
> **Please let us know if your concerns have been addressed**
>
> Dear Reviewer 2LuU,
>
> Thank you again for your review. **We hope that our rebuttal could address your questions and concerns**, such as the generalizability of DiffPano, why generating multi-view ERP panoramas is meaningful, and the downstream applications of multi-view panorama images. As the discussion phase is nearing its end, **we would be grateful to hear your feedback and wondered if you might still have any concerns we could address.**
>
> We have provided an anonymous link to the AC to forward the link to the reviewers, due to the requirements during the rebuttal stage. This anonymous link contains outdoor panorama generation and single-view vs multi-view panoramas.
>
> **It would be appreciated if you could raise your score on our paper if we address your concerns.** We thank you again for your effort in reviewing our paper.
>
> Best regards,
>
> DiffPano Authors

---

> > ### Comment · Reviewer_2LuU · 2024-08-14
> >
> > Thank you for your rebuttal. Most of my concerns have been addressed. However, I would appreciate more discussion in the revised version on the use case of multi-view panorama. For instance, while multi-view ERP cannot be directly used for VR roaming, could it enhance immersive 3D scene generation (as opposed to using just a single view)? Additionally, it would be beneficial to include more related work in panoramic 3D scene generation to further strengthen this paper, as generating panorama image is not the terminal of the real-world application.

---

> ### Author Response · Authors · 2024-08-14
> **Reply to Reviewer 2LuU**
>
> Dear Reviewer 2LuU,
>
> Thank you again for your review. We are glad that **our rebuttal has addressed your concerns**. We will **discuss more in the revised version of the use case of the multi-view panorama and include more related work(such as recent single-view panorama generation work[1, 2, 3, 4, 5, 6]) in panoramic 3D scene generation**.
>
> [1] Zhou, Haiyang, et al. "HoloDreamer: Holistic 3D Panoramic World Generation from Text Descriptions." arXiv preprint arXiv:2407.15187 (2024).
>
> [2] Li, Renjie, et al. "4K4DGen: Panoramic 4D Generation at 4K Resolution." arXiv preprint arXiv:2406.13527 (2024).
>
> [3] Wu, Tianhao, Chuanxia Zheng, and Tat-Jen Cham. "Ipo-ldm: Depth-aided 360-degree indoor rgb panorama outpainting via latent diffusion model." arXiv preprint arXiv:2307.03177 (2023).
>
> [4] Wang, Jionghao, et al. "360-degree panorama generation from few unregistered nfov images." arXiv preprint arXiv:2308.14686 (2023).
>
> [5] Lu, Zhuqiang, et al. "Autoregressive Omni-Aware Outpainting for Open-Vocabulary 360-Degree Image Generation." Proceedings of the AAAI Conference on Artificial Intelligence. Vol. 38. No. 13. 2024.
>
> [6] Wang, Hai, et al. "Customizing 360-degree panoramas through text-to-image diffusion models." Proceedings of the IEEE/CVF Winter Conference on Applications of Computer Vision. 2024.
>
>
> Compared with single-view panorama generation, **our method can generate consistent multi-view ERP panoramas, which can be transferred to the perspective view for immersive VR roaming with unlimited scapes**. There is **related work [7] on using single-view panoramas for VR roaming**. We can use this work for VR roaming.
>
> [7] DreamSpace: Dreaming Your Room Space with Text-Driven Panoramic Texture Propagation, VR2024.
>
> Additionally, we would like to **discuss the use of multi-view panoramas for interior home design**. Given a floor plan of a house, users can customize different rooms throughout the house according to their needs. Then given a textual description of the tour from different rooms (similar to the specified trajectory and textual description in our panoramic video-text pipeline), our method can be used to generate multi-view panoramic videos. In this way, users can preview their envisioned home design.
>
> More importantly, the single-view panoramic image generated by previous methods mainly supports 3DoF roaming. Our method generates multi-view panoramic images that support 6DoF roaming, serving as inputs for 360-degree Gaussian Splatting[8] or 360-degree NeRF[9, 10]. Our method also has a great potential value in relightable 360-degree NVS with the combination of 360-degree multi-view inverse rendering method [11].
>
> [8] Xiong, Haolin. SparseGS: Real-time 360° sparse view synthesis using Gaussian splatting. MS thesis. University of California, Los Angeles, 2024.
>
> [9] Huang, Huajian, et al. "360Roam: Real-Time Indoor Roaming Using Geometry-Aware ${360^\circ} $ Radiance Fields." arXiv preprint arXiv:2208.02705 (2022).
>
> [10] Chen, Zheng, et al. "PanoGRF: generalizable spherical radiance fields for wide-baseline panoramas." Advances in Neural Information Processing Systems 36 (2023): 6961-6985.
>
> [11] Li, Zhen, et al. "Multi-view inverse rendering for large-scale real-world indoor scenes." Proceedings of the IEEE/CVF Conference on Computer Vision and Pattern Recognition. 2023.
>
>
> **We hope our reply could address your questions**. As the discussion phase is nearing its end, **we would be grateful to hear your feedback and wondered if you might still have any concerns we could address.**
>
> **It would be appreciated if you could recommend accepting our paper in the reviewers-pc discussion**. We thank you again for your effort in reviewing our paper.
>
> Best regards,
>
> DiffPano Authors

---

### Author Rebuttal · Authors · 2024-08-07

We thank all reviewers for their constructive suggestions and appreciate comments like
“well-written, well-structured and easy to follow, clear" (R2LuU, R7HQE, RK8Nd), "first work, pioneer work, novel, technically sound, interesting" (R2LuU, R7HQE, RK8Nd, RYygs), "valuable, promote, enhance, useful, meaningful, facilitating" (R2LuU, R7HQE, RK8Nd, RYygs), "effectiveness, looks compact and practical to implement" (R2LuU, R7HQE, RYygs), "plausible results and visually pleasant" (R7HQE).

We hope after reading our comprehensive and detailed rebuttal, the reviewers can eliminate RK8Nd's misunderstanding about our work, change his mind, and give us a higher score. We will gladly incorporate all the feedback in the revised version.

1) The main concerns of the reviewers are the generalization (R2LuU, RK8Nd, RYygs), diversity (RYygs), and scalability (RK8Nd) of DiffPano, the left-right consistency (R7HQE) and realism (RYygs) of the generated panoramas, and the realism (RYygs) of converting the generated panoramas into perspective images. We show some generated outdoor panoramas in the PDF to verify the generalization, generate diverse panoramas with the same text description to show the diversity, and generate longer panoramic videos with image-based panorama generation to demonstrate scalability (we recommend reviewers open the video with Adobe software to see the video).

As for the left-right consistency of the generated panoramas, we propose to split the panorama into two parts and use a matching algorithm to calculate the matching pairs for verification. We don't adopt the Left-Right Consistency Error (LRCE) suggested by R7HQE, for this metric is suited in the text-to-panorama generation. Instead, we propose a Left-Right Matching Pairs (LRMP) metric to calculate the left-right consistency using matching algorithms. As suggested by R7HQE, we compare cylinder padding without data augmentation and the data augmentation method used in this paper to improve the left-right consistency. Experiments show that our simple data augmentation scheme can obtain a higher CLIP Score (CS) and Inception Score (IS), as well as lower Fréchet Inception Distance (FID). See the rebuttal to 7HQE.

We follow Text2Light to report FID and IS on panoramas to measure realism and diversity. Additionally, CLIP Score (CS) is used to evaluate the text-image consistency. While FID is widely used for image
generation, it relies on an Inception network trained on perspective images, thus less applicable for panorama images.
We adopt a variant of FID customized for panorama, Fr´echet Auto-Encoder Distance (FAED) in PanFusion,
which is used to better compare the realism. And the relevant results are as follows. Experiments show that our simple data augmentation scheme can obtain higher CLIP Score (CS) and Inception Score (IS), as well as lower Fréchet Inception Distance (FID) and FAED. See the rebuttal to 7HQE.

As suggested, we convert the generated panorama into a perspective view to further verify the realism.
Experiments show that our method achieves the lowest FID, while our method is higher than MVDiffusion in IS and slightly lower than PanFusion. It should be noted that MVDiffusion directly generates perspective images and then stitches them into panoramas. It is not in the ERP format and does not have the top and bottom parts. Our panorama generation speed is faster.

We also conduct ablation experiments on one-stage and two-stage training. Experiments show that
due to the introduction of the spherical epipolar-aware module, the multi-view consistency of the two training methods is relatively good.
Comparing the consistency is not enough to measure the difference between the two methods.
The IS of the two-stage training method is lower, and the diversity is reduced to a certain extent,
which is slightly worse than the one-stage training. However, it should be noted that the images after one-stage training will have ghosting, but the two-stage will not. We show the qualitative comparison results (the FID value of the two-stage method will also be higher). See the rebuttal to Yygs.


All in all, the panoramas generated by our method have good realism and left-right consistency with faster speed. Our method can achieve scalable, diverse, and consistent panorama generation.

2) Most importantly, Reviewer K8Nd seems to have misunderstood the core content of this paper,
the problem it solves, and the core innovation. Here, we point out the factual error. We aim to perform scalable multi-view panorama generation based on text (which can be expanded from one room to another). As far as we know, we are the first to propose text-to-multi-view panorama generation. Previous methods are limited to single-image panorama generation. The core reason is the lack of large-scale multi-view panorama datasets, and single panorama generation work cannot guarantee the multi-view consistency of the generated panorama. A new framework suitable for multi-view panorama generation is needed. To this end, we first construct a large-scale panoramic video-text dataset. Then, we design a multi-view panorama generation framework. First, we propose a diffusion model for text-to-single-view panorama generation (fine-tuned on the newly proposed panorama dataset using LoRA). To achieve multi-view consistency, we derive a spherical epipolar constraint suitable for panoramas and embed it into the network as an attention layer to realize the spherical epipolar multi-view panorama generation framework, which can achieve diverse and consistent scalable panorama generation results.

We hope that all reviewers can have in-depth discussions on K8Nd's misunderstandings to ensure that reviewer K8Nd can eliminate the factual errors and re-evaluate our work to obtain a fair review. We sincerely thank you.

3) We will polish the writing and proofread the revised version, including the typos and adding the necessary citations.

---

### Author Response · Authors · 2024-08-07
**To ACs, SACs and Reviewers for pointing out the factual error of Reviewer K8Nd**

We thank all reviewers for their constructive comments and recognition of our work ("first work, pioneer work, interesting task", "novel and technically sound", "plausible and visually pleasant panoramic results"), and ACs and SACs for their hard work.
However, Reviewer K8Nd seems to have misunderstood the core content of this paper,
the problem it solves, and the core innovation. We believe this review may be pure LLM generated.
We hope that ACs, SACs, and all reviewers can read our paper and the rebuttal and then have in-depth discussions on K8Nd's misunderstandings to ensure that reviewer K8Nd can eliminate the factual errors and re-evaluate our work to obtain a fair review.
We sincerely thank you.

Here, we point out the factual error. Reviewer K8Nd argues that our approach lacks innovation and is a technique that has been studied in previous approaches. The goal of this paper is to perform scalable multi-view panorama generation based on text (which can be expanded from one room to another). As far as we know, we are the first to propose text-to-multi-view panorama generation. Previous methods are limited to single-image panorama generation. The core reason is the lack of large-scale multi-view panorama datasets, and single-view panorama generation work cannot guarantee the multi-view consistency of the generated panorama. A new framework suitable for multi-view panorama generation is needed. To this end, we first construct a large-scale panoramic video-text dataset. Then, we design a multi-view panorama generation framework. First, we propose a diffusion model for text-to-single-view panorama generation (fine-tuned on the newly proposed panorama dataset using LoRA). To achieve multi-view consistency, we derive a spherical epipolar constraint suitable for panoramas and embed it into the network as an attention layer to realize the spherical epipolar multi-view panorama generation framework, which can achieve diverse and consistent scalable panorama generation results. Our method is different from previous methods in terms of both technique and dataset (as recognized by Reviewer 2LuU and Reviewer 7HQE), but Reviewer K8Nd did not realize the innovativeness of our technique.

We hope that our rebuttal and explanation can correct the reviewer K8Nd's misunderstanding
so that our work can be properly reviewed and obtain a satisfactory review result.
We sincerely hope that ACs and SACs can have in-depth discussions with all reviewers to review our work more fairly.

---

### Author Response · Authors · 2024-08-12
**Please comment on our rebuttal and discuss with reviewer K8Nd to correct his factual errors**

Dear Program Chairs, Senior Area Chairs, Area Chairs and Reviewers,

As the deadline for author-reviewer discussion is approaching, we **have not received any comments from the reviewer**, and **reviewer K8Nd seems to have misunderstood the core content, problems solved, and core innovations of this article**. We sincerely hope that you can **remind the reviewer to comment on our rebuttal** and **discuss with reviewer K8Nd to correct his factual errors** so that our article can receive reasonable evaluation.

Best regards,
DiffPano Authors

---

### Author Response · Authors · 2024-08-12
**Please comment on our rebuttal and discuss with reviewer K8Nd to correct his factual errors**

Dear Program Chairs, Senior Area Chairs, Area Chairs and Reviewers,

As the deadline for author-reviewer discussion is approaching, we **have not received any comments from the reviewers**, and **reviewer K8Nd seems to have misunderstood the core content, problems solved, and core innovations of this paper**. We sincerely hope that you can **remind the reviewer to comment on our rebuttal** and **discuss with reviewer K8Nd to correct his factual errors** so that our paper can receive reasonable evaluations.

Best regards, DiffPano Authors

---

### Decision · Program_Chairs · 2024-09-25

**Decision:**

Accept (poster)

**Comment:**

Reviewers are overall positive about the paper and leaning towards acceptance. The area chair also supports the decision. Nonetheless, there are many/various concerns on the submission. Please read the comments/concerns carefully and prepare the camera ready.